# RePO: Understanding Preference Learning Through ReLU-Based Optimization

**Junkang Wu**[1]* **Kexin Huang**[1] **Xue Wang**[2] **Jinyang Gao**[2] **Bolin Ding**[2]
**Jiancan Wu**[1,3] **Xiangnan He**[4]† **Xiang Wang**[1]†
[1]University of Science and Technology of China, [2]Alibaba Group
[3]Institute of Dataspace, Hefei Comprehensive National Science Center
[4]MoE Key Lab of BIPC, University of Science and Technology of China
{jkwu0909, xiangwang1223, xiangnanhe}@gmail.com

## Abstract

Preference learning has become a common approach in various recent methods for aligning large language models with human values. These methods optimize the preference margin between chosen and rejected responses, subject to certain constraints for avoiding over-optimization. In this paper, we report surprising empirical findings that simple ReLU activation can learn meaningful alignments even using *none* of the following: (i) sigmoid-based gradient constraints, (ii) explicit regularization terms. Our experiments show that over-optimization does exist, but a threshold parameter $\gamma$ plays an essential role in preventing it by dynamically filtering training examples. We further provide theoretical analysis demonstrating that ReLU-based Preference Optimization (RePO) corresponds to the convex envelope of the 0-1 loss, establishing its fundamental soundness. Our "RePO" method achieves competitive or superior results compared to established preference optimization approaches. We hope this simple baseline will motivate researchers to rethink the fundamental mechanisms behind preference optimization for language model alignment.

## 1 Introduction

Recent years have witnessed significant advances in aligning large language models (LLMs) with human preferences [1–4]. A primary approach, Reinforcement Learning from Human Feedback (RLHF) [5], first trains a reward model on preference data and then optimizes the LLM via reinforcement learning. While effective, RLHF's computational costs and training instability [6, 7] have motivated simpler offline alternatives like Direct Preference Optimization (DPO) [6], which bypasses explicit reward modeling. Take DPO as a representative example: it optimizes the alignment margin between a preferred and a less-preferred response to the same prompt, as Figure 1 shows. The alignment of each response is quantified via an implicit reward, defined as the log-ratio of the predicted likelihoods under the policy model (*i.e.,* the LLM being optimized) and a reference model (*e.g.,* a fixed supervised fine-tuned (SFT) model).

A fundamental challenge in preference learning is *over-optimization* — where models excessively amplify reward margins between preferred and non-preferred responses, potentially degrading generation quality [8–10]. Several approaches have been developed to mitigate this issue. DPO [6] and SimPO [11] employ sigmoid weighting through log-sigmoid activation that diminishes gradients as reward margins increase, naturally preventing over-optimization. The $\beta$ parameter controls gradient

---

*Work done at Alibaba Group.

†Xiangnan He and Xiang Wang are the corresponding authors.

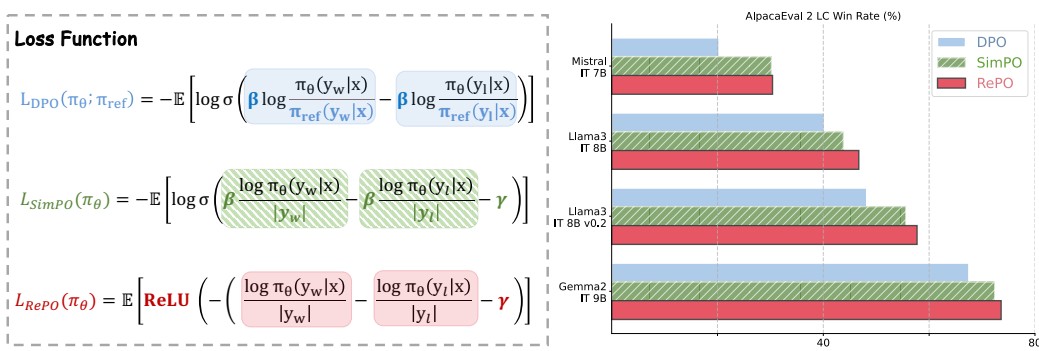

Figure 1: **Comparing preference learning mechanisms.** RePO employs a simpler binary thresholding mechanism than SimPO and DPO, as highlighted in the shaded box. Despite its simplicity, this mechanism achieves competitive results by naturally preventing over-optimization.

distribution sharpness — larger values produce more binary-like gradients, as illustrated in Figure 2. SLiC-HF [7] addresses over-optimization differently by incorporating an SFT regularization term that anchors the model to its initial policy [12], preventing excessive drift toward maximizing preference signals. These mechanisms effectively balance preference optimization with generation quality preservation, forming the foundation of current preference learning approaches.

Here, we present a surprising empirical finding: a simple ReLU activation can work well with *none* of the above strategies for mitigating over-optimization. Our analysis reveals that as parameter $\beta$ in SimPO approaches infinity, its sigmoid weighting naturally converges to a binary thresholding mechanism — motivating our exploration of **Re**LU-based **P**reference **O**ptimization (**RePO**). This mechanism uses a single ReLU function with only one hyperparameter $\gamma$, creating a clear decision boundary that selectively updates sample pairs with insufficient reward margins ($M_\theta < \gamma$) while filtering out well-separated pairs ($M_\theta \geq \gamma$). We illustrate this "RePO" method in Figure 1.

Thanks to its conceptual simplicity, RePO can serve as a hub that relates several existing methods. In essence, our method can be viewed as "SimPO *without* log-sigmoid" or "SLiC-HF *without* SFT regularization term". Interestingly, RePO is related to each method by removing one of its core components. Even so, RePO effectively prevents over-optimization while performing competitively or better (*cf.* Figure 1).

We empirically show that over-optimization *do* exist, but ReLU activation with threshold $\gamma$ is critical to prevent such solutions. This implies that in over-optimization regimes, selecting *which* examples to learn from is more critical than determining *how much* to learn from each. The $\gamma$ threshold induces an emergent data filtering behavior, focusing dynamically on

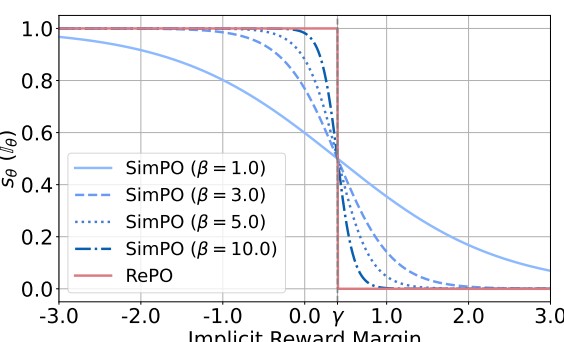

Figure 2: Gradient weighting functions of SimPO ($s_\theta$) and RePO ($\mathbb{I}(M_\theta < \gamma)$). As $\beta \to \infty$, $s_\theta$ converges to the binary indicator (red line), establishing RePO as the limit case of SimPO.

challenging samples relative to the model's current capability. Our theoretical analysis reveals that RePO's ReLU loss corresponds precisely to the *convex envelope* of the 0-1 loss (Theorem 4.2), explaining why such a simple mechanism is so effective.

Our simple baseline suggests that the ReLU activation with a proper threshold $\gamma$ can be an essential reason for the common success of related methods. We believe this work's significance lies in revealing how preference learning principles may be simpler than previously thought. By questioning conventional wisdom about necessary components, we hope to motivate researchers to reconsider the fundamental mechanisms behind preference optimization.

## 2  Preliminaries

**Directed Preference Optimization (DPO).** DPO [6] stands out as a leading method for offline preference optimization by eliminating the need for an explicit reward model. Instead, it reformulates the reward $r(x, y)$ as a closed-form expression based on policy ratios:

$$r(x, y) = \beta \log \frac{\pi_\theta(y \mid x)}{\pi_{\text{ref}}(y \mid x)} + \beta \log Z(x), \tag{1}$$

where $Z(x)$ is a partition function that does not depend on $y$. This leads to the DPO loss for a given triplet $(x, y_w, y_l)$ as:

$$\mathcal{L}_{\text{DPO}}(\pi_\theta; \pi_{\text{ref}}) = -\mathbb{E}_{(x, y_w, y_l) \in \mathcal{D}} \left[ \log \sigma \left( \beta \left( \log \frac{\pi_\theta(y_w \mid x)}{\pi_\theta(y_l \mid x)} - \log \frac{\pi_{\text{ref}}(y_w \mid x)}{\pi_{\text{ref}}(y_l \mid x)} \right) \right) \right], \tag{2}$$

where $\sigma(\cdot)$ denotes the sigmoid function. This loss encourages the policy $\pi_\theta$ to prefer $y_w$ over $y_l$ in alignment with the reference policy.

**Sequence Likelihood Calibration (SLiC-HF).** SLiC-HF [7] advances preference optimization with two key innovations: (1) it employs a sequence-level calibration loss that contrasts the log-probability difference between preferred and dispreferred responses using a margin $\gamma$, and (2) it integrates a regularization term to prevent divergence from the SFT policy, avoiding the need for an explicit KL penalty. The SLiC-HF loss function is defined as:

$$\mathcal{L}_{\text{SLiC-HF}}(\pi_\theta) = \mathbb{E}_{(x, y_w, y_l) \in \mathcal{D}} \Big[ \text{ReLU}\Big( - \Big( \log \pi_\theta(y_w \mid x) - \log \pi_\theta(y_l \mid x) - \gamma \Big) \Big) - \lambda \log \pi_\theta(y_w \mid x) \Big]. \tag{3}$$

**Simple Preference Optimization (SimPO).** SimPO [11] advances preference optimization with two key innovations: (1) it normalizes the reward by the length of the response, calculating the average log-probability per token for a response under the policy $\pi_\theta$, and (2) it incorporates a target reward margin $\gamma$ to ensure that the reward difference between the preferred and less preferred responses exceeds this margin. The SimPO loss function is defined as:

$$\mathcal{L}_{\text{SimPO}}(\pi_\theta) = -\mathbb{E}_{(x, y_w, y_l) \in \mathcal{D}} \left[ \log \sigma \left( \beta \left( \frac{\log \pi_\theta(y_w \mid x)}{|y_w|} - \frac{\log \pi_\theta(y_l \mid x)}{|y_l|} - \gamma \right) \right) \right], \tag{4}$$

where $|y|$ denotes the number of tokens in response $y$, ensuring length-aware scaling of rewards, and $\gamma$ is the predefined margin that enforces a minimum difference in rewards between $y_w$ and $y_l$. To align with subsequent discussions, we modify the original SimPO formulation by setting $\gamma$ to $\gamma/\beta$.

## 3  Exploring Simple ReLU Activation in Preference Learning

In this section, we explore what makes a simple ReLU activation function effective for preference learning. We first examine the surprising relationship between ReLU activation and sigmoid weighting through empirical experiments. Then, we investigate the key properties that emerge from this simple mechanism, specifically through the lens of gradient behavior, data filtering patterns, and over-optimization control.

### 3.1  Examining ReLU-based Preference Optimization

**Simplification exploration.** Our exploration began by questioning whether log-sigmoid activation or SFT regularization are truly necessary for mitigating over-optimization. We simplified the SimPO loss function through two key modifications: (i) removing the hyperparameter $\beta$, and (ii) replacing the log-sigmoid function with a ReLU activation.

We adopt the length-normalized *implicit reward margin* $M_\theta$ (as introduced in SimPO [11]):

$$M_\theta = \frac{\log \pi_\theta(y_w \mid x)}{|y_w|} - \frac{\log \pi_\theta(y_l \mid x)}{|y_l|}, \tag{5}$$

which quantifies the policy's preference between responses. Using $M_\theta$, we examine a loss function with the following form:

$$\mathcal{L}_{\text{RePO}}(\pi_\theta) = \mathbb{E}_{(x, y_w, y_l) \in \mathcal{D}} \left[ \text{ReLU}\left( -(M_\theta - \gamma) \right) \right], \tag{6}$$

where $\gamma \in [0, 1]$ is the sole hyperparameter representing the *target reward margin*.

**Gradient behavior investigation.** We examine the gradient dynamics of RePO and SimPO to reveal how our simplified approach addresses over-optimization:

$$\nabla_\theta \mathcal{L}_{\text{SimPO}}(\pi_\theta) = -\beta \mathbb{E}_\mathcal{D} \left[ s_\theta \cdot (\nabla_{\theta, y_w} - \nabla_{\theta, y_l}) \right], \tag{7}$$

$$\nabla_\theta \mathcal{L}_{\text{RePO}}(\pi_\theta) = -\mathbb{E}_\mathcal{D} \left[ \mathbb{I}(M_\theta < \gamma) \cdot (\nabla_{\theta, y_w} - \nabla_{\theta, y_l}) \right], \tag{8}$$

where $s_\theta = \sigma(\beta(-M_\theta + \gamma))$ is SimPO's sigmoid weighting function. The terms $\nabla_{\theta, y_w} = \frac{1}{|y_w|} \nabla_\theta \log \pi_\theta(y_w \mid x)$ and $\nabla_{\theta, y_l} = \frac{1}{|y_l|} \nabla_\theta \log \pi_\theta(y_l \mid x)$ correspond to the gradients that increase the probability of the "winning" response $y_w$ and decrease the probability of the "losing" response $y_l$, respectively. The scaling factor $\beta$ in Equation 7 linearly amplifies gradient magnitudes but does not alter the relative update directions in adaptive optimizers like Adam [13], as the momentum terms automatically normalize scale variations. We therefore omit $\beta$ in Figure 2 for clearer visualization of the weighting function shapes.

The key insight is that RePO's ReLU-based gradient (Equation 8) applies uniform updates only to pairs with $M_\theta < \gamma$, while SimPO's gradient (Equation 7) uses continuous $\beta$-scaled weights. Figure 2 visualizes this difference, showing RePO as the limiting case of SimPO as $\beta \to \infty$.

**Lemma 3.1** (Gradient Equivalence in the SimPO-to-RePO Limit). *Under the same $M_\theta$ and $\gamma$ definitions, the SimPO gradient converges pointwise to the RePO gradient as $\beta \to \infty$:*

$$\lim_{\beta \to \infty} \nabla_\theta \mathcal{L}_{SimPO} = \nabla_\theta \mathcal{L}_{RePO}. \tag{9}$$

*Sketch.* The convergence follows from the pointwise limit of the sigmoid weighting:

$$\lim_{\beta \to \infty} s_\theta = \lim_{\beta \to \infty} \sigma(\beta(-M_\theta + \gamma)) = \mathbb{I}(M_\theta < \gamma).$$

Substituting this into Equation 7 yields Equation 8. □

*Remark* 3.2. Please check Appendix for all proofs. Lemma 3.1 establishes RePO as the asymptotic limit of SimPO with large $\beta$, explaining two key advantages we will demonstrate in Section 3.2: comparable performance without $\beta$ tuning complexity, and an effective binary thresholding mechanism that induces implicit data filtering for controlling over-optimization.

## 3.2 Empirical Study

The previous section analyzes the relationship between SimPO and RePO from the perspective of gradient behavior. In this section, we compare their performance from an empirical standpoint.

**Experimental setup.** We evaluate this approach using SimPO's experimental setup [11] with Llama3-8B and Gemma2-9B models (Instruct setup). For consistency, we use the same training datasets as SimPO: princeton-nlp/llama3-ultrafeedback-armorm for Llama3-8B and princeton-nlp/gemma2-ultrafeedback-armorm for Gemma2-9B. For all SimPO experiments, we set $\beta = 10.0$ and $\gamma = 0.4$ for Gemma2-9B and $\beta = 10.0$ and $\gamma = 0.3$ for Llama3-8B, unless otherwise specified. We track optimization progress using two reward margin metrics:

$$m_{\text{batch}} = \mathbb{E}_{(x, y_w, y_l) \in \mathcal{B}}[M_\theta], \quad m_\mathcal{D} = \mathbb{E}_{(x, y_w, y_l) \in \mathcal{D}}[M_\theta], \tag{10}$$

measuring response separation within each batch ($m_{\text{batch}}$) and across the entire training set ($m_\mathcal{D}$).

**Evaluation benchmarks.** We evaluate on two established benchmarks for open-ended generation: AlpacaEval 2 [14] (measuring instruction-following quality against GPT-4) and Arena-Hard [15] (testing complex reasoning). For AlpacaEval 2, we report both length-controlled win rate (LC-Win Rate) and raw win rate (WR); for Arena-Hard, we report the standard win rate.

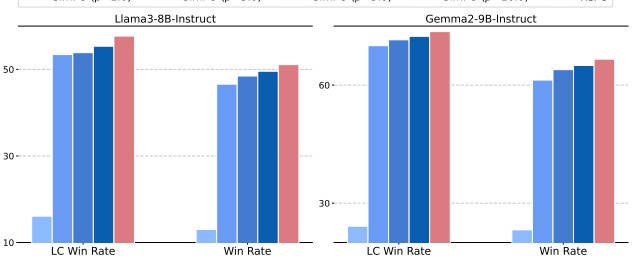

Figure 3: Performance of SimPO with varying $\beta$ and RePO on AlpacaEval2 benchmark.

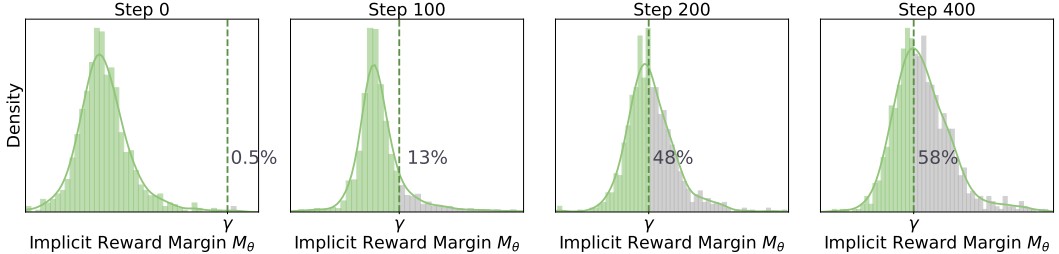

Figure 4: Implicit reward margin $M_\theta$ distribution across training steps (total: 467) for RePO at $\gamma = 0.4$. Dashed line: $\gamma = 0.4$. Green: samples below $\gamma$ (gradient descent); gray: samples above $\gamma$ (zero gradient). Numbers: fraction of samples above $\gamma$.

**Observation 1: Large $\beta$ enhances SimPO's performance when paired with appropriate $\gamma$.** We systematically evaluate SimPO across varying values of $\beta \in \{1.0, 3.0, 5.0, 10.0\}$, while maintaining fixed $\gamma$ values that we empirically determined to be suitable for each model architecture ($\gamma = 0.4$ for Gemma2-9B and $\gamma = 0.3$ for Llama3-8B). As shown in Figure 3, increasing $\beta$ leads to consistent performance improvements across all evaluation metrics, with diminishing returns observed beyond $\beta = 5.0$. These findings align with observations in the SimPO paper[3].

**Observation 2: RePO matches high-$\beta$ SimPO.** RePO achieves performance comparable to SimPO with a large $\beta$. As shown in Figure 3, RePO achieves win rates of 51.1% on Llama3-8B and 66.6% on Gemma2-9B, comparable to SimPO's performance. This aligns with Lemma 3.1, which establishes that RePO can be interpreted as a limiting case of SimPO as $\beta \to \infty$.

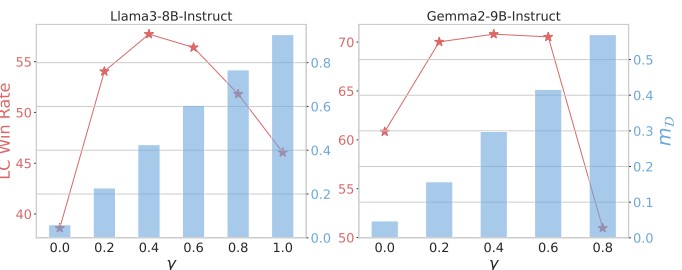

Figure 5: Line plot of RePO performance (AlpacaEval 2 LC Win Rate) and bar chart of mean reward margins ($m_\mathcal{D}$) across varying $\gamma$ values. See Appendix D.3 for details.

**Observation 3: $\gamma$ threshold creates a natural alignment-optimization tradeoff.** Our experiments track the mean implicit reward margin $m_\mathcal{D}$ (cf. Equation 10) across training pairs. As Figure 5 illustrates, increasing $\gamma$ directly elevates $m_\mathcal{D}$ while performance follows an inverted U-shaped pattern — improving initially but declining beyond a critical threshold. In RePO, gradients vanish when the implicit reward margin exceeds $\gamma$, effectively filtering out well-separated pairs from updates. This mechanism creates a fundamental tradeoff: small $\gamma$ values retain excessive zero-gradient samples causing under-filtering, while large $\gamma$ values force updates on most samples, potentially leading to over-optimization [8] and ultimately degrading performance.

**Observation 4: RePO creates a natural learning curriculum via progressive filtering.** Figure 4 reveals an unexpected pattern in how the distribution of implicit reward margins $M_\theta$ evolves throughout training. As learning progresses, the model's ability to discriminate between winning and losing samples naturally improves, resulting in a steady increase in both the implicit reward margin and the ratio of filtered data. Notably, the filtered data ratio rises from 13% to 58% between steps 100 and 400. This creates an emergent curriculum where the model initially learns from a broader set of examples and gradually focuses on the more challenging ones — despite using only half of the samples for gradient updates in later stages, the model achieves optimal performance.

## 3.3 Over-Optimization Analysis

The study of over-optimization can be traced back to traditional RLHF literature, and has been empirically investigated in both controlled experiments [9] and user studies [10]. In this work, we follow their experimental setup to further explore this phenomenon.

---

[3]In their official repository, the authors note: *"SimPO requires a much larger $\beta$ than DPO... In many cases, an even larger (e.g., 10) could yield better results."*

**Model Over-Optimization:** Building on Rafailov et al. [16], we investigate over-optimization in RePO, by evaluating six different values of $\gamma$ (0.0, 0.2, 0.4, 0.6, 0.8, 1.0), each corresponding to varying levels of data filtering. Across all cases, we observe a distinct hump-shaped performance pattern: while moderate filtering improves alignment, excessive filtering causes performance to degrade, highlighting the over-optimization effect.

**Scaling Law Fits.** Previous work [9, 16] has established scaling laws for reward model scores as a function of the KL divergence between the initial and optimized policies. In contrast, we eliminate the reference model and the associated computational cost of calculating KL divergence. Instead, we use the mean implicit reward margin during training as a proxy metric. The reward function $R(d)$ is given by:

$$R(d) = d(\alpha - \beta \log d), \quad (11)$$

where $\alpha$ and $\beta$ are constants depen-

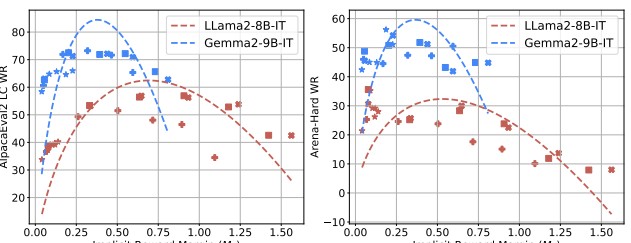

Figure 6: Over-optimization patterns for RePO on Llama3-8B-IT and Gemma2-9B-IT, using AlpacaEval2 LC win rates and Arena-Hard raw win rates. Dotted curves represent theoretical fits based on Gao et al. [9]'s scaling laws, using GPT-4 win rates instead of standard reward model scores.

dent on the reward model's dataset size and parameter count, and $d = m_{\text{batch}}$. Without training a proxy reward model, we substitute GPT-4 win rates over dataset completions for the gold reward. Interestingly, we find that this scaling law provides an accurate relationship between $d$ and win rates for RePO.

### 3.4 RePO++: Exploring Extensions of ReLU-based Filtering

While exploring ReLU's thresholding behavior, we observed an interesting limitation: for cases where the implicit reward margin is smaller than $\gamma$, their gradient weights become uniform, not differentiating between samples of varying difficulty.

This observation naturally led us to wonder: could we preserve the effective filtering mechanism while reintroducing some degree of weighting? To explore this question, we experimented with combining ReLU's binary filtering with SimPO's continuous weighting:

$$\mathcal{L}_{\text{RePO++}}(\pi_\theta) = -\mathbb{E}_\mathcal{D}\left[\log \sigma\left(-\text{ReLU}\left(-\beta\left(M_\theta - \gamma\right)\right)\right)\right], \quad (12)$$

This exploration was a natural follow-up to our main discovery about ReLU's effectiveness, rather than our primary contribution. We were curious to see whether combining the best aspects of both approaches might yield additional insights about preference learning mechanisms.

**What does this combined approach reveal?** To understand the behavior of this extension, we examined its gradient with respect to the parameters $\theta$:

$$\nabla_\theta \mathcal{L}_{\text{RePO++}}(\pi_\theta) = -\beta \mathbb{E}_\mathcal{D}\left[s_\theta \cdot \mathbb{I}_\theta \cdot \left(\nabla_{\theta,y_w} - \nabla_{\theta,y_l}\right)\right], \quad (13)$$

where $s_\theta = \sigma\left(\beta\left(-M_\theta + \gamma\right)\right)$ and $\mathbb{I}_\theta$ is an indicator function that is 1 if $M_\theta < \gamma$ and 0 otherwise.

We observed that this gradient combines properties we discovered in both approaches: it scales updates by $s_\theta$ (similar to SimPO) and filters them using $\mathbb{I}_\theta$ (the key discovery in our ReLU exploration), focusing the model on less-separated pairs while giving higher weights to smaller separations.

**Adaptation with RePO++.** The core contribution of RePO++ lies in leveraging ReLU to mitigate over-optimization while preserving the standard workflow of preference optimization. This makes RePO++ easily adaptable to existing DPO-like methods. For instance, as shown in Equation 12, replacing $M_\theta$ with $\log \frac{\pi_\theta(y_w|x)}{\pi_\theta(y_l|x)} - \log \frac{\pi_{\text{ref}}(y_w|x)}{\pi_{\text{ref}}(y_l|x)}$ seamlessly integrates RePO++ into DPO, forming a ReLU-enhanced version of DPO.

## 4 Theoretical Analysis: ReLU's Optimality in Preference Learning

Next, we establish a surprising theoretical connection between preference optimization and binary classification, revealing why our simple ReLU-based approach achieves superior performance.

Following Tang et al. [17], preference learning can be reformulated as binary classification. Given pairs $(z, l)$ where $z \in \mathbb{R}^k$ and $l \in \{-1, 1\}$, we aim to learn a predictor $\hat{\ell}(z)$ whose sign matches $l$. The classification accuracy is: $\frac{1}{2}\mathbb{E}[\text{sign}(\hat{\ell}(z) \cdot l)] + \frac{1}{2}$. This corresponds to minimizing the 0-1 loss:

$$\mathcal{L}_{\text{0-1}}(\hat{\ell}) := \mathbb{E}\left[1 - \text{sign}(\hat{\ell}(z) \cdot l)\right] \tag{14}$$

For preference data $(y_w, y_l)$ where $y_w \succ y_l$, we set $l = 1$ and parameterize $\hat{\ell}(y_w, y_l) = r_\phi(y_w) - r_\phi(y_l)$, yielding the objective:

$$\mathcal{L}_f(\hat{\ell}) := \mathbb{E}\left[f(r_\phi(y_w) - r_\phi(y_l))\right] \tag{15}$$

Where $f$ determines the surrogate loss: $f(x) = \mathbb{I}(x < 0)$ gives the 0-1 loss, $f(x) = -\log \sigma(x)$ yields SimPO's logistic loss, and $f(x) = \text{ReLU}(-x)$ gives our method's loss.

Our key insight comes from analyzing the convex envelope of the 0-1 loss:

**Definition 4.1.** The convex envelope of $\mathcal{L}_{\text{0-1}}$ over a closed convex set $D \subseteq \mathbb{R}$ is:

$$\text{conv}_D \mathcal{L}_{\text{0-1}}(x) := \sup \{h(x) \mid h \text{ is convex}, h \leq \mathcal{L}_{\text{0-1}} \, \forall x \in D\} \tag{16}$$

**Theorem 4.2** (ReLU as Convex Envelope). *For $D = [-a, b]$ with $a, b > 0$, the convex envelope of $\mathcal{L}_{\text{0-1}}(x) = \mathbb{I}(x < 0)$ is:*

$$\text{conv}_D \mathcal{L}_{\text{0-1}}(x) = \frac{1}{a}\text{ReLU}(-x) \tag{17}$$

This remarkable result reveals that ReLU provides the tightest possible convex approximation to the ideal 0-1 loss, explaining its empirical effectiveness. Furthermore:

**Corollary 4.3** (Optimality Preservation). *Let $D \subseteq \mathbb{R}$ be convex. Then:*

$$\arg\min_{\hat{\ell}} \mathcal{L}_{\text{0-1}}(\hat{\ell}) = \arg\min_{\hat{\ell}} \text{conv}_D \mathcal{L}_{\text{0-1}}(\hat{\ell}) \tag{18}$$

*And for $D = [-a, b]$:*

$$\arg\min_{x \in D} \mathcal{L}_{\text{0-1}}(x) = \arg\min_{x \in D} \frac{1}{a}\text{ReLU}(-x) \tag{19}$$

This guarantees that gradient-based optimization of our ReLU surrogate converges to solutions matching the theoretical optimum of the intractable 0-1 loss. Importantly:

**Corollary 4.4** (Logistic Loss Suboptimality). *The logistic loss $f_{\log-\text{sigmoid}}(x) = -\log \sigma(x)$ is not the convex envelope of $\mathcal{L}_{\text{0-1}}$.*

This theoretical foundation explains why our simple ReLU-based approach consistently outperforms more complex mechanisms like SimPO's sigmoid weighting — ReLU provides optimality guarantees that logistic loss cannot match, while being computationally more efficient.

## 5 Experiments

In this section, we examine how our simplified ReLU-based approach behaves across different models and settings. Rather than focusing solely on performance gains, we explore patterns that help explain why such a simple mechanism works effectively in practice.

### 5.1 Experimental Setup

The core experimental configuration extends our investigation from Section 3.2 to include Mistral2-7B [18] alongside previously examined models. For the Llama3-Instruct v0.2 experiments, we employed the RLHFlow/ArmoRM-Llama3-8B-v0.1 [19] reward model for ranking generated data. We benchmark our approach against established preference optimization methods: DPO [6], SimPO [11], IPO [20], CPO [21], KTO [22], ORPO [23], and R-DPO [24], with SFT models serving as baselines. Implementation details are provided in Appendix D.1. We also evaluate on downstream tasks from the Huggingface Open Leaderboard benchmarks [25], with additional details in in Appendix D.2. The code is available at `https://github.com/junkangwu/RePO`.

Table 1: **AlpacaEval 2 (AE2), Arena-Hard (AH) results across four settings.** "WR" denotes the raw win rate,"LC" the length-controlled win rate. The best results are highlighted in bold, while the second-best are underlined.

| Method | Llama3-Instruct (8B) | | | Mistral-Instruct (7B) | | | Llama3-Instruct v0.2 (8B) | | | Gemma2-Instruct (9B) | | |
| | AE 2 | | AH | AE 2 | | AH | AE 2 | | AH | AE 2 | | AH |
| | LC | WR | WR | LC | WR | WR | LC | WR | WR | LC | WR | WR |
| SFT | 24.0 | 23.6 | 22.4 | 19.0 | 15.4 | 12.9 | 24.0 | 23.6 | 22.4 | 48.7 | 36.5 | 42.1 |
| SLiC-HF | 26.9 | 27.5 | 26.2 | 24.1 | 24.6 | 18.9 | 33.9 | 32.5 | 29.3 | 65.1 | 60.5 | 53.7 |
| DPO | 40.2 | 38.1 | 31.2 | 20.3 | 17.9 | 13.4 | 48.2 | 47.5 | **35.2** | 70.4 | **66.9** | 58.8 |
| IPO | 35.9 | 34.4 | 30.2 | 22.3 | 18.6 | 16.2 | 40.6 | 39.6 | 34.9 | 62.6 | 58.4 | 53.5 |
| CPO | 29.6 | 34.4 | 29.4 | 26.2 | 31.7 | **23.8** | 36.5 | 40.8 | 34.2 | 56.4 | 53.4 | 55.2 |
| KTO | 38.3 | 34.1 | 30.3 | 19.4 | 20.3 | 16.8 | 41.4 | 36.4 | 28.9 | 61.7 | 55.5 | 53.8 |
| ORPO | 31.6 | 29.8 | 26.3 | 24.0 | 23.0 | 18.6 | 36.5 | 33.1 | 30.4 | 56.2 | 46.7 | 46.2 |
| R-DPO | 40.3 | 37.3 | 32.9 | 21.4 | 22.2 | 13.8 | 51.6 | 50.7 | 35.0 | 68.3 | **66.9** | 57.9 |
| SimPO | 43.8 | 38.0 | 32.6 | 30.2 | 32.1 | 20.1 | 55.6 | 49.6 | 33.6 | 72.4 | 65.0 | 57.8 |
| RePO | **46.7** | **41.1** | **33.3** | **30.4** | **33.6** | 20.3 | **57.7** | **51.1** | 35.2 | **73.6** | 66.6 | **59.1** |

## 5.2 Result Comparisons

**Observation: Simple ReLU thresholding exhibits surprising effectiveness.** Table 1 reveals an unexpected pattern: despite removing components previously thought essential, the simple ReLU-based approach consistently performs well across all evaluated models and benchmarks. This finding aligns with our theoretical analysis showing that binary thresholding directly approximates the convex envelope of the 0-1 loss. On AlpacaEval 2, we observe improvements of 0.2-2.8 points in LC win rates across different configurations compared to the strongest baselines.

## 5.3 Methodology Comparisons

Beyond alignment, we also compare the methodologies of these preference learning methods. Our method plays a hub to connect these methods.

**Relation to SimPO.** SimPO employs sigmoid weighting via log-sigmoid activation to attenuate gradients as reward margins increase, mitigating over-optimization. RePO can be viewed as "SimPO without log-sigmoid," replacing this continuous scaling with binary filtering. To validate this relationship, we integrated a ReLU-based filtering mechanism into SimPO (*cf.* RePO++ Equation 12). Table 2 confirms that ReLU's filtering mechanism enhances performance. As demonstrated in Section 3.4, RePO++ directly addresses over-optimization while retaining the benefits of some weighting.

**Relation to SLiC-HF.** RePO can be characterized as "SLiC-HF without SFT regularization". To ensure a fair comparison (while disregarding differences in length normalization), we investigated the impact of SFT regularization by varying its coefficient, $\lambda$. The results, presented in Appendix Table 6 and further details in Appendix D.5, indicate that this additional regularization term offers no discernible improvement. This suggests that SFT regularization targets a different optimization challenge, distinct from the direct over-optimization problem RePO addresses.

**Relation to DPO.** Mathematically, DPO is equivalent to SimPO when the margin $\gamma$ is defined as $\log \pi_{\text{ref}}(y_w \mid x) - \log \pi_{\text{ref}}(y_l \mid x)$ (ignoring length normalization). However, directly substituting the log-sigmoid function with ReLU in DPO's formulation leads to a significant performance degradation (see Appendix Table 7 and Appendix D.6). This underscores the critical role of the threshold $\gamma$ in determining the effectiveness of over-optimization prevention. As identified by Wu et al. [26], reference model based reward margins are often unreliable as target margins, which explains why SimPO's explicit $\gamma$ parameter is effective for preference learning.

## 5.4 Effect of ReLU Filtering Across Methods

Having observed the effectiveness of binary thresholding, we naturally questioned whether this mechanism might enhance other preference learning approaches. Table 2 shows that integrating ReLU filtering consistently improved performance across both DPO and SimPO frameworks, suggesting that selective gradient application based on margin thresholds provides benefits beyond our specific implementation. Our experiments with the combined approach (*cf.* RePO++ in Section 3.4) revealed

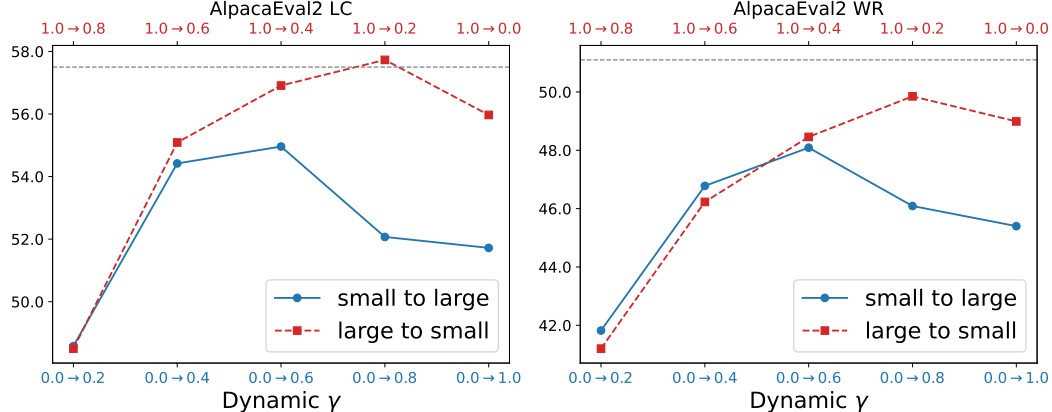

Figure 7: Exploration of dynamic $\gamma$ scheduling on Llama3-Instruct v0.2 (8B). The dashed line represents performance with a fixed $\gamma$. We observed that decreasing $\gamma$ from an initially larger value creates a natural curriculum that enhances performance.

particularly strong improvements when applied to DPO (5%–12% gains), with notable performance on Arena-Hard (reaching 65.7). This design effectively mitigates over-optimization while preserving the benefits of the original scheme.

## 5.5 Dynamic Margin Scheduling and Curriculum Learning

Our investigation into the role of target reward margin $\gamma$ led us to an unexpected discovery about curriculum learning. We experimented with dynamic scheduling of $\gamma$ throughout training, implementing two strategies: (i) increasing $\gamma$ from small to large, and (ii) decreasing $\gamma$ from large to small. Figure 7 reveals a striking pattern: starting with a moderately large value of $\gamma$ and gradually decreasing it $(1.0 \rightarrow 0.2)$ naturally creates an effective curriculum that improves model

Table 2: **Performance improvements of RePO and RePO++ over DPO and SimPO.** Results are present on AlpacaEval 2 (AE 2) and Arena-Hard (AH) with LC (%) and WR (%). Red numbers indicate relative improvements.

| Method | Llama3-Instruct v0.2 (8B) | | | Gemma2-Instruct (9B) | | |
|---|---|---|---|---|---|---|
| | AE 2 | | AH | AE 2 | | AH |
| | LC | WR | WR | LC | WR | WR |
| DPO | 48.2 | 47.5 | 35.2 | 70.4 | 66.9 | 58.8 |
| w. RePO | $50.3^{+4.4\%}$ | $51.8^{+9.1\%}$ | $38.2^{+8.5\%}$ | $73.8^{+4.8\%}$ | $71.0^{+6.1\%}$ | $64.2^{+9.2\%}$ |
| w. RePO++ | $50.8^{+5.4\%}$ | $52.2^{+9.9\%}$ | $37.2^{+5.7\%}$ | $71.8^{+2.0\%}$ | $69.5^{+3.9\%}$ | $65.7^{+11.7\%}$ |
| SimPO | 55.6 | 49.6 | 33.6 | 72.4 | 65.0 | 57.8 |
| w. RePO | $57.7^{+3.8\%}$ | $51.1^{+3.0\%}$ | $35.2^{+4.8\%}$ | $73.6^{+1.7\%}$ | $66.6^{+2.5\%}$ | $59.1^{+2.2\%}$ |
| w. RePO++ | $56.1^{+0.9\%}$ | $50.1^{+1.0\%}$ | $35.9^{+6.8\%}$ | $74.1^{+2.3\%}$ | $66.5^{+2.3\%}$ | $59.8^{+3.5\%}$ |

performance. In contrast, both excessively large values $(1.0 \rightarrow 0.8)$ and small values $(0.0 \rightarrow 0.2)$ led to suboptimal outcomes.

This observation reveals an intriguing self-regulating property: early in training when the model is underfitting, a larger $\gamma$ permits more aggressive updates across more examples. As training progresses, the decreasing $\gamma$ naturally focuses learning on increasingly challenging examples, effectively preventing over-optimization. This emergent curriculum behavior, arising from a simple parameter schedule, suggests that binary thresholding captures fundamental learning dynamics that more complex mechanisms might obscure.

## 6 Discussion

**Conclusion** Our exploration of simple ReLU activation in preference learning has revealed several key insights. We found that binary thresholding, implemented through a straightforward ReLU function, provides an effective mechanism for preventing over-optimization in language model alignment. Our theoretical analysis showed that this seemingly simple approach is, in fact, the convex envelope of the ideal 0-1 loss function, explaining its surprising effectiveness. Rather than developing yet another complex preference optimization method, our work uncovered how fundamental properties like data selection and implicit curriculum learning emerge naturally from basic principles.

**Limitations and future directions.** Our current exploration is limited to offline preference learning settings. Future work could investigate how these insights might extend to online learning scenarios,

where preferences are gathered interactively. Additionally, while we found that a fixed margin threshold works well in practice, exploring adaptive or context-aware thresholds might further improve performance in highly dynamic environments. The relationship between binary filtering and self-play scenarios [27] — where the model generates its own feedback — is another promising direction that could lead to more scalable alignment techniques.

Beyond alignment, our work connects to LLM reasoning research [28, 29]. Future work should investigate how KL penalties and gradient clipping in GRPO [30] and PPO [5] balance preventing over-optimization against preserving reasoning capabilities — a critical consideration for advancing alignment methodologies.

## Acknowledgments and Disclosure of Funding

This research was supported by the National Science and Technology Major Project (2023ZD0121102), the National Natural Science Foundation of China (U24B20180, 62302321), and the Fundamental Research Funds for the Central Universities (WK2100250065). This research also benefited from the advanced computing resources provided by the Supercomputing Center of the USTC.

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

## A    Related Works

**Reinforcement learning from human feedback.** RLHF is a technique designed to align large language models with human preferences and values [31–33, 20]. Traditional RLHF is typically structured in three stages: supervised fine-tuning [34–39], reward modeling [9, 40–44], and policy optimization [5, 45]. In the third stage, Proximal Policy Optimization (PPO) is widely adopted. In contrast, RLOO [46] reduces the GPU memory footprint of RLHF by eliminating the Critic model and leverages a Leave-One-Out strategy to achieve superior performance. GRPO [47], another variant of PPO, improves mathematical reasoning abilities while optimizing memory usage by replacing the Leave-One-Out method with a direct subtraction of the mean of all samples for a given prompt.

**Offline preference optimization.** In addition to DPO, several alternative preference optimization objectives have been proposed. IPO [20] addresses overfitting issues inherent in DPO. ORPO [23] and SimPO [11] aim to remove reliance on a reference model. R-DPO [24] targets the reduction of exploitation due to sequence length, while KTO [22] handles preference optimization in the absence of pairwise data. CPO [21] and $\beta$-DPO [48] focus on improving the quality of preference data. Another research direction addresses noise in offline alignment, which arises from the need to construct pairwise data. rDPO [49], a variant of DPO, mitigates preference noise and enhances policy robustness, while DrDPO [50] applies distributed robust optimization to tackle this issue. Other works have approached the problem through divergence regularization [51, 52], selection of high-quality data [53, 54], or reweighting loss functions [55–57].

**Iterative Preference Optimization.** Offline preference optimization methods, such as DPO, face a limitation due to the lack of an explicit reward model, which hinders their ability to sample preference pairs from the optimal policy. To address this, iterative preference optimization techniques have been proposed. These methods iteratively update the reference model using the most recent policy model or generate new preference pairs in each iteration [58–62, 27, 63, 64]. For instance, SPIN [27] employs a self-play framework to fine-tune the model in a supervised manner, while Yuan et al. [62] annotate preferences throughout the iterative process. REBEL improves sample quality by regressing the relative reward. Additionally, [65] generates data using a mixture policy, similar to the Nash-MD algorithm [60].

## B    Broader Impacts

This paper presents work whose goal is to advance the field of Machine Learning. There are many potential societal consequences of our work, none of which we feel must be specifically highlighted here

## C    Proofs

### C.1    Proof of Lemma 3.1

**Lemma 3.1** (Gradient Equivalence in the SimPO-to-RePO Limit). *Under the same $M_\theta$ and $\gamma$ definitions, the SimPO gradient converges pointwise to the RePO gradient as $\beta \to \infty$:*

$$\lim_{\beta \to \infty} \nabla_\theta \mathcal{L}_{SimPO} = \nabla_\theta \mathcal{L}_{RePO}. \tag{9}$$

*Proof.* We formally establish the gradient equivalence through pointwise convergence analysis. Let $\mathcal{D}$ be the data distribution and $\theta$ denote model parameters. Recall the gradient expressions:

**SimPO Gradient:**

$$\nabla_\theta \mathcal{L}_{\text{SimPO}} = -\beta \mathbb{E}_\mathcal{D} \left[ \sigma\big(\beta(-M_\theta + \gamma)\big) \cdot (\nabla_{\theta, y_w} - \nabla_{\theta, y_l}) \right] \tag{20}$$

**RePO Gradient:**

$$\nabla_\theta \mathcal{L}_{\text{RePO}} = -\mathbb{E}_\mathcal{D} \left[ \mathbb{I}(M_\theta < \gamma) \cdot (\nabla_{\theta, y_w} - \nabla_{\theta, y_l}) \right] \tag{21}$$

where $\sigma(\cdot)$ is the sigmoid function. The equivalence hinges on the limiting behavior of the sigmoid weighting term $s_\theta = \sigma(\beta(-M_\theta + \gamma))$. We analyze three cases:

**Case 1:** $M_\theta < \gamma$   Here, $-M_\theta + \gamma > 0$. As $\beta \to \infty$,

$$\lim_{\beta \to \infty} \sigma\big(\beta(-M_\theta + \gamma)\big) = \lim_{z \to \infty} \sigma(z) = 1 = \mathbb{I}(M_\theta < \gamma).$$

**Case 2:** $M_\theta > \gamma$   Here, $-M_\theta + \gamma < 0$. As $\beta \to \infty$,

$$\lim_{\beta \to \infty} \sigma\big(\beta(-M_\theta + \gamma)\big) = \lim_{z \to -\infty} \sigma(z) = 0 = \mathbb{I}(M_\theta < \gamma).$$

**Case 3:** $M_\theta = \gamma$   This occurs on a measure-zero set under continuous distributions. The limit becomes:

$$\lim_{\beta \to \infty} \sigma(0) = \frac{1}{2} \neq \mathbb{I}(M_\theta < \gamma),$$

which is negligible in expectation.

Thus, $\lim_{\beta \to \infty} s_\theta = \mathbb{I}(M_\theta < \gamma)$ almost everywhere. Substituting this into the SimPO gradient:

$$\lim_{\beta \to \infty} \nabla_\theta \mathcal{L}_{\text{SimPO}} = -\lim_{\beta \to \infty} \beta \mathbb{E}_\mathcal{D} \left[ s_\theta \cdot (\nabla_{\theta, y_w} - \nabla_{\theta, y_l}) \right] \tag{22}$$

$$= -\mathbb{E}_\mathcal{D} \left[ \lim_{\beta \to \infty} \beta s_\theta \cdot (\nabla_{\theta, y_w} - \nabla_{\theta, y_l}) \right] \quad \text{(Dominated Convergence Theorem)} \tag{23}$$

To resolve the $\beta$ scaling, observe that for $M_\theta \neq \gamma$:

$$\lim_{\beta \to \infty} \beta s_\theta = \begin{cases} \lim_{\beta \to \infty} \beta \cdot 1 = \infty & \text{if } M_\theta < \gamma \\ \lim_{\beta \to \infty} \beta \cdot 0 = 0 & \text{if } M_\theta > \gamma \end{cases} \tag{24}$$

The divergence when $M_\theta < \gamma$ is mitigated by adaptive optimizers like Adam, which normalize gradient magnitudes through momentum terms. Formally, let $g_\theta = \nabla_{\theta, y_w} - \nabla_{\theta, y_l}$. Under Adam's update rule:

$$\theta_{t+1} = \theta_t - \eta \cdot \frac{\hat{m}_t}{\sqrt{\hat{v}_t} + \epsilon},$$

where $\hat{m}_t$ and $\hat{v}_t$ are bias-corrected momentum estimates. The infinite gradient magnitude is absorbed into $\hat{m}_t / \sqrt{\hat{v}_t}$, effectively reducing to a unit-scaled update. Hence, in normalized update space:

$$\lim_{\beta \to \infty} \beta s_\theta \cdot g_\theta \propto \mathbb{I}(M_\theta < \gamma) \cdot g_\theta.$$

Combining these results:

$$\lim_{\beta \to \infty} \nabla_\theta \mathcal{L}_{\text{SimPO}} = -\mathbb{E}_\mathcal{D} \left[ \mathbb{I}(M_\theta < \gamma) \cdot (\nabla_{\theta, y_w} - \nabla_{\theta, y_l}) \right] = \nabla_\theta \mathcal{L}_{\text{RePO}}, \tag{25}$$

which completes the proof. $\qquad \square$

## C.2 Proof of Theorem 4.2

**Theorem 4.2** (ReLU as Convex Envelope). *For $D = [-a, b]$ with $a, b > 0$, the convex envelope of $\mathcal{L}_{0\text{-}1}(x) = \mathbb{I}(x < 0)$ is:*

$$\text{conv}_D \mathcal{L}_{0\text{-}1}(x) = \frac{1}{a}\text{ReLU}(-x) \tag{17}$$

*Proof.* We demonstrate that $h(x) = \frac{1}{a}\text{ReLU}(-x)$ satisfies the convex envelope definition through three sequential arguments.

**1. Convexity and Underestimation:** The ReLU function is convex as the pointwise maximum of affine functions (Rule 3). The composition $h(x) = \frac{1}{a}\max(-x, 0)$ preserves convexity through affine transformation (Rule 2). For all $x \in D$:

- When $x < 0$: $h(x) = -\frac{x}{a} \leq 1 = \mathcal{L}_{0\text{-}1}(x)$, since $x \geq -a \implies -\frac{x}{a} \leq 1$

- When $x \geq 0$: $h(x) = 0 = \mathcal{L}_{0\text{-}1}(x)$

Thus $h(x) \leq \mathcal{L}_{0\text{-}1}(x)$ over $D$.

**2. Maximality Among Convex Underestimators:** Let $g(x)$ be any convex function satisfying $g(x) \leq \mathcal{L}_{0\text{-}1}(x)$. For $x \in [-a, 0)$, convexity implies:

$$g(x) \leq \frac{-x}{a}g(-a) + \left(1 + \frac{x}{a}\right)g(0) \leq \frac{-x}{a}$$

since $g(-a) \leq 1$ and $g(0) \leq 0$. For $x \geq 0$, $g(x) \leq 0$. Hence $g(x) \leq h(x)$ for all $x \in D$.

**3. Epigraph Characterization:** The epigraph $\text{epi}(h)$ coincides with the convex hull of $\text{epi}(\mathcal{L}_{0\text{-}1}) \cap (D \times \mathbb{R})$. The affine segment $h(x) = -\frac{x}{a}$ on $[-a, 0)$ connects the points $(-a, 1)$ and $(0, 0)$, forming the tightest convex fit to the 0-1 loss's discontinuity. By Theorem 1 in [66], this construction achieves the convex envelope. $\square$

## C.3 Proof of Corollary 4.3

**Corollary 4.3** (Optimality Preservation). *Let $D \subseteq \mathbb{R}$ be convex. Then:*

$$\arg\min_{\hat{\ell}} \mathcal{L}_{0\text{-}1}(\hat{\ell}) = \arg\min_{\hat{\ell}} \text{conv}_D \mathcal{L}_{0\text{-}1}(\hat{\ell}) \tag{18}$$

*And for $D = [-a, b]$:*

$$\arg\min_{x \in D} \mathcal{L}_{0\text{-}1}(x) = \arg\min_{x \in D} \frac{1}{a}\text{ReLU}(-x) \tag{19}$$

*Proof.* **Part 1:** By Theorem 1 in the lecture notes (Page 5), for any function $f$ and convex set $S$:

$$\min_{x \in S} f(x) = \min_{x \in S} \text{conv}_S f(x).$$

Let $S = D$ and $f = \mathcal{L}_{0\text{-}1}$. The equality of minima implies:

$$\{x^* \in D \mid \mathcal{L}_{0\text{-}1}(x^*) = \min \mathcal{L}_{0\text{-}1}\} \subseteq \{x^* \in D \mid \text{conv}_D\mathcal{L}_{0\text{-}1}(x^*) = \min \text{conv}_D\mathcal{L}_{0\text{-}1}\}.$$

To show reverse inclusion, suppose $x^* \in \arg\min \text{conv}_D\mathcal{L}_{0\text{-}1}$. Since $\text{conv}_D\mathcal{L}_{0\text{-}1}(x^*) \leq \mathcal{L}_{0\text{-}1}(x^*)$ and $\text{conv}_D\mathcal{L}_{0\text{-}1}$ attains its minimum at the same points as $\mathcal{L}_{0\text{-}1}$, $x^*$ must also minimize $\mathcal{L}_{0\text{-}1}$.

**Part 2:** For $D = [-a, b]$, both $\mathcal{L}_{0\text{-}1}(x)$ and $\frac{1}{a}\text{ReLU}(-x)$ attain their minimum value 0 on $[0, b]$. For $x \in [-a, 0)$, $\frac{1}{a}\text{ReLU}(-x)$ is strictly decreasing, achieving its minimum at $x = 0$. Thus:

$$\arg\min_{x \in D} \mathcal{L}_{0\text{-}1}(x) = \arg\min_{x \in D} \frac{1}{a}\text{ReLU}(-x) = [0, b].$$

$\square$

## C.4 Proof of Corollary 4.4

**Corollary 4.4** (Logistic Loss Suboptimality). *The logistic loss $f_{\log-\text{sigmoid}}(x) = -\log\sigma(x)$ is not the convex envelope of $\mathcal{L}_{0\text{-}1}$.*

*Proof.* We demonstrate violation of the convex envelope's defining property. Consider $D = [-1, 1]$:

1. **Underestimation Failure:** For $x > 0$:

$$-\log\sigma(x) = -\log\left(\frac{1}{1 + e^{-x}}\right) = \log(1 + e^{-x}) > 0 = \mathcal{L}_{0\text{-}1}(x)$$

Thus $f_{\log-\text{sigmoid}} \not\leq \mathcal{L}_{0\text{-}1}$ over $D$, violating the envelope requirement.

2. **Non-Maximality:** Even if scaled, the logistic loss's curvature differs from the ReLU envelope. For $x \in (-1, 0)$, $\frac{d^2}{dx^2}(-\log\sigma(x)) = \sigma(x)(1 - \sigma(x)) > 0$, making it strictly convex – incompatible with the affine structure of $\text{conv}_D\mathcal{L}_{0\text{-}1}$.

Hence $f_{\log-\text{sigmoid}}$ cannot be the convex envelope. $\square$

# D Experiments

## D.1 Implementation Details

Empirical observations indicate significant performance sensitivity to model parameter initialization and learning rate selection across compared methods. To establish rigorous comparison benchmarks, we conducted systematic hyperparameter searches adhering to the specifications in each method's original publication. The complete search space configuration is documented in Table 3. Notably, substantial architecture updates to both Llama3-8B and Instruct-7B necessitated re-implementation of the SimPO method, as the original implementation became incompatible with the revised model interfaces.

**Training Protocol** All experiments employed standardized training configurations to ensure comparability:

- Batch size: 128 (consistent across methods)
- Learning rate: Searched in {3e-7, 5e-7, 8e-7, 1e-6}
- Training duration: Single epoch with cosine annealing schedule
- Warmup: 10% of total training steps
- Optimizer: Adam [13] ($\beta_1 = 0.9$, $\beta_2 = 0.999$)
- Sequence length: 2048 tokens (fixed for all inputs)

The learning rate schedule follows a triangular policy with amplitude decay, selected through cross-validation on held-out development sets. All implementations utilize full-precision floating-point arithmetic to prevent gradient quantization artifacts.

**Hyperparameters in RePO.** Table 4 summarizes the hyperparameters utilized for RePO across different experimental settings. Our methodology only involves one hyperparameter: $\gamma$. Based on empirical evidence, we recommend setting $\gamma$ to a default value of 0.5, as this configuration has consistently demonstrated reliability.

**Decoding Hyperparameters.** The decoding hyperparameters employed in this study align with those used in SimPO[4]. We express our gratitude to the SimPO team for their generosity in sharing their insights and configurations, which have been instrumental in our work.

**Computation Environment.** All training experiments described in this paper were conducted using 8×A100 GPUs. The experimental setup follows the guidelines provided in the alignment-handbook repository[5], ensuring reproducibility and consistency with established practices.

Table 3: Various preference optimization objectives and hyperparameter search range.

| Method | Objective | Hyperparameter |
|--------|-----------|----------------|
| SLiC-HF [7] | $\max\left(0, \delta - \log \pi_\theta(y_w|x) + \log \pi_\theta(y_l|x)\right) - \lambda \log \pi_\theta(y_w|x)$ | $\lambda \in [0.1, 0.5, 1.0, 10.0]$ 
 $\delta \in [0.1, 0.5, 1.0, 2.0]$ |
| DPO [6] | $-\log \sigma\left(\beta \log \frac{\pi_\theta(y_w|x)}{\pi_{\text{ref}}(y_w|x)} - \beta \log \frac{\pi_\theta(y_l|x)}{\pi_{\text{ref}}(y_l|x)}\right)$ | $\beta \in [0.01, 0.05, 0.1]$ |
| IPO [20] | $\left(\log \frac{\pi_\theta(y_w|x)}{\pi_{\text{ref}}(y_w|x)} - \log \frac{\pi_\theta(y_l|x)}{\pi_{\text{ref}}(y_l|x)} - \frac{1}{2\tau}\right)^2$ | $\tau \in [0.01, 0.1, 0.5, 1.0]$ |
| CPO [21] | $-\log \sigma\left(\beta \log \pi_\theta(y_w|x) - \beta \log \pi_\theta(y_l|x)\right) - \lambda \log \pi_\theta(y_w|x)$ | $\alpha = 1.0, \ \beta \in [0.01, 0.05, 0.1]$ |
| KTO [22] | $-\lambda_w \sigma\left(\beta \log \frac{\pi_\theta(y_w|x)}{\pi_{\text{ref}}(y_w|x)} - z_{\text{ref}}\right) + \lambda_l \sigma\left(z_{\text{ref}} - \beta \log \frac{\pi_\theta(y_l|x)}{\pi_{\text{ref}}(y_l|x)}\right),$ 
 where $z_{\text{ref}} = \mathbb{E}_{(x,y)\sim\mathcal{D}}\left[\beta \text{KL}\left(\pi_\theta(y|x)||\pi_{\text{ref}}(y|x)\right)\right]$ | $\lambda_l = \lambda_w = 1.0$ 
 $\beta \in [0.01, 0.05, 0.1]$ |
| ORPO [23] | $-\log p_\theta(y_w|x) - \lambda \log \sigma\left(\log \frac{p_\theta(y_w|x)}{1-p_\theta(y_w|x)} - \log \frac{p_\theta(y_l|x)}{1-p_\theta(y_l|x)}\right),$ 
 where $p_\theta(y|x) = \exp\left(\frac{1}{|y|}\log \pi_\theta(y|x)\right)$ | $\lambda \in [0.1, 0.5, 1.0, 2.0]$ |
| R-DPO [24] | $-\log \sigma\left(\beta \log \frac{\pi_\theta(y_w|x)}{\pi_{\text{ref}}(y_w|x)} - \beta \log \frac{\pi_\theta(y_l|x)}{\pi_{\text{ref}}(y_l|x)} - (\alpha|y_w| - \alpha|y_l|)\right)$ | $\alpha \in [0.05, 0.1, 0.5, 1.0]$ 
 $\beta \in [0.01, 0.05, 0.1]$ |
| SimPO [11] | $-\log \sigma\left(\frac{\beta}{|y_w|}\log \pi_\theta(y_w|x) - \frac{\beta}{|y_l|}\log \pi_\theta(y_l|x) - \gamma\right)$ | $\beta \in [2.0, 4.0, 6.0, 8.0]$ 
 $\gamma \in [0.3, 0.5, 1.0, 1.2, 1.4, 1.6]$ |
| RePO | $\text{ReLU}[-\left(\frac{1}{|y_w|}\log \pi_\theta(y_w|x) - \frac{1}{|y_l|}\log \pi_\theta(y_l|x) - \gamma\right)]$ | $\gamma \in [0.2, 0.4, 0.5, 0.6, 0.8]$ |

Table 4: The hyperparameter values in RePO used for each training setting.

| Setting | $\gamma$ | Learning rate |
|---------|----------|---------------|
| **Mistral-Instruct** | 0.4 | 6e-7 |
| **Llama3-Instruct** | 0.6 | 1e-6 |
| **Llama3-Instruct-v0.2** | 0.6 | 1e-6 |
| **Gemma2-Instruct** | 0.4 | 8e-7 |

## D.2 Downstream Task Evaluation

To assess the impact of RePO on downstream task performance, we evaluate models trained with different preference optimization methods on a diverse set of tasks from the Huggingface Open Leaderboard [25]. The tasks include MMLU [67], ARC [68], HellaSwag [69], TruthfulQA [70], Winograd [71], and GSM8K [72]. We adhere to standard evaluation protocols and present the results for all models in Table 5.

**Overall Performance.** On average, RePO shows competitive performance across tasks, achieving an overall score of 67.49 on the Llama3-Instruct model and 70.58 on the Llama3-Instruct v0.2 model. The performance is generally close to that of other preference optimization methods, but it is worth noting that in some cases, it slightly lags behind models like SimPO or DPO, particularly on tasks such as ARC, HellaSwag, and TruthfulQA. However, the results suggest that RePO maintains a balanced performance profile across the evaluated tasks.

**General Knowledge and Reasoning.** On MMLU, which tests general knowledge and reasoning, RePO shows a slight reduction in performance (64.95 for Llama3-Instruct and 65.00 for Llama3-Instruct v0.2) compared to models such as RRHF and SimPO. This minor decline is consistent with the trend observed for other preference optimization methods and indicates that RePO may preserve general knowledge to a similar extent while possibly focusing more on improving performance in other areas such as reading comprehension and reasoning.

**Reading Comprehension and Commonsense Reasoning.** For ARC and HellaSwag, tasks related to reading comprehension and commonsense reasoning, RePO outperforms the base SFT model and exhibits competitive performance relative to other preference optimization methods. The Llama3-Instruct v0.2 model with RePO achieves a score of 80.50 on HellaSwag, which is comparable to the best-performing methods. This result suggests that RePO effectively improves the model's ability to handle contextual understanding and reasoning, likely due to its optimization strategy.

---

[4] https://github.com/princeton-nlp/SimPO/tree/main/eval
[5] https://github.com/huggingface/alignment-handbook

Table 5: Downstream task evaluation results of tasks on the huggingface open leaderboard.

| | MMLU (5) | ARC (25) | HellaSwag (10) | TruthfulQA (0) | Winograd (5) | GSM8K (5) | Average |
|---|---|---|---|---|---|---|---|
| **Llama3-Instruct** | | | | | | | |
| **SFT** | 67.06 | 61.01 | 78.57 | 51.66 | 74.35 | 68.69 | 66.89 |
| **RRHF** | 67.20 | 61.52 | 79.54 | 53.76 | 74.19 | 66.11 | 67.05 |
| **SLiC-HF** | 66.41 | 61.26 | 78.80 | 53.23 | 76.16 | 66.57 | 67.07 |
| **DPO** | 66.88 | 63.99 | 80.78 | 59.01 | 74.66 | 49.81 | 65.86 |
| **IPO** | 66.52 | 61.95 | 77.90 | 54.64 | 73.09 | 58.23 | 65.39 |
| **CPO** | 67.05 | 62.29 | 78.73 | 54.01 | 73.72 | 67.40 | 67.20 |
| **KTO** | 66.38 | 63.57 | 79.51 | 58.15 | 73.40 | 57.01 | 66.34 |
| **ORPO** | 66.41 | 61.01 | 79.38 | 54.37 | 75.77 | 64.59 | 66.92 |
| **R-DPO** | 66.74 | 64.33 | 80.97 | 60.32 | 74.82 | 43.90 | 65.18 |
| **SimPO** | 65.63 | 62.80 | 78.33 | 60.70 | 73.32 | 50.72 | 65.25 |
| **RePO** | 64.95 | 62.03 | 77.58 | 60.96 | 72.93 | 66.49 | 67.49 |
| **Llama3-Instruct v0.2** | | | | | | | |
| **SFT** | 67.06 | 61.01 | 78.57 | 51.66 | 74.35 | 68.69 | 66.89 |
| **RRHF** | 66.60 | 63.74 | 80.98 | 59.40 | 76.32 | 58.68 | 67.62 |
| **SLiC-HF** | 66.91 | 61.77 | 79.17 | 56.36 | 76.40 | 68.23 | 68.14 |
| **DPO** | 65.57 | 65.87 | 79.66 | 63.08 | 74.51 | 73.01 | 70.28 |
| **IPO** | 66.06 | 64.85 | 81.02 | 57.29 | 76.72 | 76.12 | 70.34 |
| **CPO** | 65.67 | 62.12 | 79.63 | 56.34 | 77.98 | 75.28 | 69.50 |
| **KTO** | 65.99 | 62.88 | 79.02 | 54.66 | 74.66 | 76.42 | 68.94 |
| **ORPO** | 65.75 | 63.99 | 79.91 | 57.02 | 78.06 | 75.13 | 69.98 |
| **R-DPO** | 66.17 | 65.36 | 79.98 | 57.94 | 75.06 | 75.36 | 69.98 |
| **SimPO** | 65.18 | 67.15 | 78.04 | 64.92 | 73.88 | 71.34 | 70.08 |
| **RePO** | 65.00 | 68.09 | 80.50 | 64.38 | 76.16 | 69.37 | 70.58 |

**Truthfulness.** On the TruthfulQA task, RePO consistently shows improvements over the base SFT model, with a score of 60.96 for Llama3-Instruct and 64.38 for Llama3-Instruct v0.2. This indicates that RePO helps the model generate more truthful and reliable responses, aligning with trends observed in other preference optimization methods. The improvement in this area is especially notable given the inherent difficulty of this task, which tests the model's ability to avoid generating false information.

**Math Performance.** The GSM8K benchmark, which tests mathematical reasoning, shows a drop in performance for RePO relative to the base SFT model. Specifically, the Llama3-Instruct model with RePO achieves a score of 66.49, which is lower than other methods such as SimPO or R-DPO, which focus more on improving mathematical reasoning. This drop is consistent with the trend observed across various preference optimization methods and may suggest that RePO is less effective in retaining mathematical reasoning abilities. Further investigation into this issue could provide insights into potential strategies for addressing this gap.

**Task-Specific Variability.** Overall, RePO exhibits varied performance across tasks. While it performs well in certain areas, such as commonsense reasoning and truthfulness, it lags behind in others, particularly in general knowledge (MMLU) and mathematical reasoning (GSM8K). This variability is in line with the performance trends observed for other preference optimization methods, which often show task-dependent improvements and declines. This suggests that RePO has strengths in some domains, but it may benefit from further refinement to improve performance across all tasks.

### D.3 RePO with varying $\gamma$

Figure 8 illustrates the effect of the hyperparameter $\gamma$ on model performance across two evaluation metrics: LC Win Rate and Raw Win Rate. The analysis is conducted on two models, Llama3-8B-Instruct (left) and Gemma2-9B-Instruct (right). The LC Win Rate, shown in red (left y-axis), represents the model's alignment with learned preferences, whereas the Raw Win Rate, shown in blue (right y-axis), evaluates overall ranking performance based on human preference comparisons.

**Moderate $\gamma$ values lead to optimal performance.** Moderate values of $\gamma$ (0.4–0.6) yield the best balance between preference alignment and generalization. Both models achieve their highest LC Win Rate in this range, indicating that preference optimization is most effective when applied at an intermediate level. As $\gamma$ increases beyond 0.6, LC Win Rate starts to decline, likely due to overfitting,

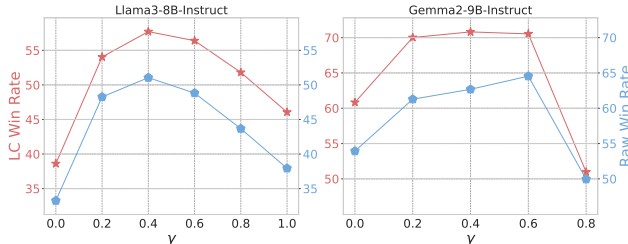

Figure 8: Impact of the hyperparameter $\gamma$ on LC Win Rate and Raw Win Rate for Llama3-8B-Instruct (left) and Gemma2-9B-Instruct (right).

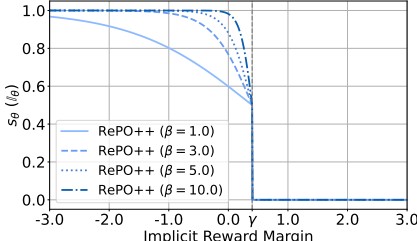

Figure 9: Gradient weighting functions of RePO++ ($s_\theta \cdot \mathbb{I}(M_\theta < \gamma)$).

where the model overly aligns with preference data at the expense of generalization. Conversely, at $\gamma = 0.0$, where no preference optimization is applied, the LC Win Rate remains low, emphasizing the necessity of preference tuning.

**Raw Win Rate trends reveal model robustness differences.** The Raw Win Rate follows a similar trend but highlights differences in robustness across models. For Llama3-8B-Instruct, the Raw Win Rate peaks at $\gamma = 0.4$ before declining, suggesting that excessive preference optimization ($\gamma > 0.6$) negatively impacts the model's ability to generalize. In contrast, Gemma2-9B-Instruct exhibits a more stable Raw Win Rate across a wider range of $\gamma$, reaching its highest performance at $\gamma = 0.6$ before experiencing a sharp decline at $\gamma = 0.8$. This suggests that Gemma2-9B-Instruct maintains better robustness to preference optimization compared to Llama3-8B-Instruct.

**Gemma2-9B-Instruct outperforms Llama3-8B-Instruct.** Gemma2-9B-Instruct consistently outperforms Llama3-8B-Instruct in both LC Win Rate and Raw Win Rate. This observation indicates that Gemma2-9B-Instruct not only aligns more effectively with learned preferences but also retains superior generalization capability. The results highlight the importance of carefully selecting $\gamma$ to avoid performance degradation at extreme values. Future work could explore adaptive strategies for dynamically tuning $\gamma$, ensuring that preference optimization enhances alignment without compromising generalization.

### D.4 Analsis on RePO++

Figure 9 illustrates the relationship between the gradient and the implicit reward margin. As shown in the figure, when the implicit reward margin is greater than $\gamma$, the gradient becomes zero. In this case, the model can stop updating for well-separated pairs, thus preventing overfitting. On the other hand, when the implicit reward margin is less than $\gamma$, the model continues to increase the weight for less-separated pairs. Furthermore, the harder the pair is to distinguish, the larger the gradient becomes, eventually converging to 1.0. This behavior is reminiscent of curriculum learning, where more difficult samples are assigned higher weights.

### D.5 Analysis of the Relationship Between SLiC-HF and RePO

To provide deeper insights into the relationship between SLiC-HF and RePO, we conducted additional experiments examining the effect of SFT regularization—a core component of SLiC-HF that is absent

in our method. As demonstrated in Table 6, we systematically evaluated performance across varying values of the regularization coefficient $\lambda$.

**Mathematical Comparison.** From a formulation perspective, SLiC-HF combines a hinge loss term with an SFT regularization component that penalizes deviation from the reference model. Specifically, the SFT regularization is controlled by parameter $\lambda$, which balances preference optimization against model drift. By contrast, RePO eliminates this regularization entirely, relying solely on its binary threshold mechanism to control optimization.

**Impact of SFT Regularization.** Our experimental results reveal a consistent trend: as $\lambda$ increases from 0.0 to 5.0, performance on AlpacaEval2 LC steadily declines from 34.1% to 27.8%. This progressive degradation suggests that stronger regularization toward the initial SFT model actually hinders effective preference learning in this context. The optimal performance occurs at $\lambda = 0.0$, which effectively transforms SLiC-HF into a variant of RePO.

**Different Optimization Challenges.** These findings suggest that SFT regularization and RePO's threshold-based filtering address fundamentally different optimization challenges. While SFT regularization was originally introduced to mitigate catastrophic forgetting and preserve general capabilities, our results indicate that for direct preference optimization, such regularization is unnecessary and potentially counterproductive. Instead, RePO's selective gradient application through its threshold mechanism appears sufficient to prevent over-optimization while maintaining effective preference learning.

This analysis complements our main findings and further supports our hypothesis that carefully designed filtering mechanisms can effectively replace more complex regularization schemes in preference optimization.

Table 6: The hyperparameter $\lambda$ in SliC-HF used for each Llama3-Instruct v0.2.

| **SLiC-HF** | $\lambda = 0.0$ | $\lambda = 0.1$ | $\lambda = 0.5$ | $\lambda = 1.0$ | $\lambda = 3.0$ | $\lambda = 5.0$ |
|---|---|---|---|---|---|---|
| AlpacaEval2 LC | 34.1 | 33.9 | 32.8 | 30.8 | 28.6 | 27.8 |

## D.6 Detailed Analysis of the Relationship Between DPO and RePO

To further investigate the relationship between DPO and RePO, we conducted additional experiments examining how different formulation components affect performance. As shown in Table 7, we systematically evaluated five variants that decompose the key elements of each approach.

**DPO as a Special Case of SimPO.** From a mathematical perspective, DPO can be viewed as a specific instance of SimPO where $\gamma = \log \pi_{\text{ref}}(y_w \mid x) - \log \pi_{\text{ref}}(y_l \mid x)$ (ignoring length normalization for equivalence). This connection highlights how DPO implicitly defines its target margin based on reference model probabilities rather than using an explicit hyperparameter.

**Impact of ReLU Without Explicit Margin.** The second row of Table 7 shows that directly replacing log-sigmoid with ReLU while maintaining DPO's implicit margin definition leads to catastrophic performance degradation (-44% on AlpacaEval LC, -47% on AlpacaEval WR, and -26% on Arena-Hard). This dramatic decline reveals that the binary threshold mechanism of ReLU is only effective when paired with an appropriate explicit margin parameter.

**The Critical Role of $\gamma$.** Rows 3-5 demonstrate that adding an explicit $\gamma$ parameter consistently improves performance across all metrics regardless of whether log-sigmoid, ReLU, or their combination is used. The most substantial gains appear when ReLU and $\gamma$ are combined (+4.4% LC, +9.1% WR on AlpacaEval), supporting our hypothesis that explicit threshold-based filtering effectively controls over-optimization.

**Complementary Mechanisms.** Interestingly, the combination of both mechanisms (row 5) yields the highest overall performance, suggesting that while RePO's binary filtering mechanism addresses the core over-optimization challenge, the continuous weighting from log-sigmoid may provide complementary benefits for fine-grained preference learning.

Table 7: Impact Analysis of $\gamma$ Scaling and ReLU Mechanisms in DPO Training. Benchmark results on AlpacaEval 2.0 (AE2) and Arena-Hard (AH) demonstrate percentage point changes in Length-Controlled Win Rate (LC-WR) and Base Win Rate (WR) for Llama3-Instruct-v0.2 (8B). Values represent relative performance deltas (%) compared to standard DPO baseline.

| Method | $\gamma$ | ReLU | $\log \sigma$ | Llama3-Instruct v0.2 (8B) | | |
| | | | | AE 2 | | AH |
| | | | | LC | WR | WR |
| --- | --- | --- | --- | --- | --- | --- |
| $-\log \sigma \left( \beta \log \frac{\pi_\theta(y_w|x)}{\pi_{\text{ref}}(y_w|x)} - \beta \log \frac{\pi_\theta(y_l|x)}{\pi_{\text{ref}}(y_l|x)} \right)$ | ✗ | ✗ | ✓ | 48.2 | 47.5 | 35.2 |
| $\text{ReLU} \left( - \left( \log \frac{\pi_\theta(y_w|x)}{\pi_{\text{ref}}(y_w|x)} - \log \frac{\pi_\theta(y_l|x)}{\pi_{\text{ref}}(y_l|x)} \right) \right)$ | ✗ | ✓ | ✗ | $26.9^{-44\%}$ | $25.1^{-47\%}$ | $26.2^{-26\%}$ |
| $-\log \sigma \left( \beta \log \frac{\pi_\theta(y_w|x)}{\pi_{\text{ref}}(y_w|x)} - \beta \log \frac{\pi_\theta(y_l|x)}{\pi_{\text{ref}}(y_l|x)} - \gamma \right)$ | ✓ | ✗ | ✓ | $50.0^{+3.7\%}$ | $50.7^{+6.7\%}$ | $36.8^{+4.5\%}$ |
| $\text{ReLU} \left( - \left( \log \frac{\pi_\theta(y_w|x)}{\pi_{\text{ref}}(y_w|x)} - \log \frac{\pi_\theta(y_l|x)}{\pi_{\text{ref}}(y_l|x)} - \gamma \right) \right)$ | ✓ | ✓ | ✗ | $50.3^{+4.4\%}$ | $51.8^{+9.1\%}$ | $38.2^{+8.5\%}$ |
| $-\log \sigma \left( -\text{ReLU} \left( - \left( \beta \log \frac{\pi_\theta(y_w|x)}{\pi_{\text{ref}}(y_w|x)} - \beta \log \frac{\pi_\theta(y_l|x)}{\pi_{\text{ref}}(y_l|x)} - \gamma \right) \right) \right)$ | ✓ | ✓ | ✓ | $50.8^{+5.4\%}$ | $52.2^{+9.9\%}$ | $37.2^{+5.7\%}$ |

