# OpenReview forum: "RePO: Understanding Preference Learning Through ReLU-Based Optimization"
_NeurIPS.cc/2025/Conference — NeurIPS 2025 poster_

### Official Review · Reviewer_uBRY · 2025-06-30

**Clarity:** 3
**Significance:** 4
**Originality:** 2
**Rating:** 5
**Confidence:** 4

**Summary:**

The paper introduces ReLU-based Preference Optimization (RePO), a simplified approach to preference optimization for aligning LLMs with human preferences. Unlike established methods such as SimPO, DPO, and SLiC-HF, which rely on sigmoid-based weighting or explicit regularization to prevent over-optimization, RePO leverages a ReLU-based loss with a single margin threshold hyperparameter $\gamma$.
This threshold-based mechanism acts as a binary filter that dynamically and selectively updates only those training samples whose reward margins fall below the threshold. The key insight is that RePO is the asymptotic limit of SimPO as its scaling factor β approaches infinity, making RePO a conceptually intuitive but theoretically grounded approach.

Extensive experiments on instruction-following and reasoning benchmarks, including AlpacaEval 2 and Arena-Hard, show that RePO outperform or match strong baselines like DPO and SimPO across several model families. The authors also performed experiment on diverse aspect for further understanding of the proposed method,  identifying an implicit curriculum learning behavior induced by the filtering mechanism and showing how dynamic scheduling of $\gamma$ can further improve performance. The authors further propose RePO++, which blends ReLU filtering with sigmoid-based weighting to combine the strengths of both methods.
With this empirical studies, the paper argues that simple threshold-based data selection may be more critical than fine-grained gradient control in preference learning.

**Questions:**

- Could the authors consider including at least one controlled experiment under the replicated conditions from prior work (e.g. Gao et al.) to increase reproducibility and practical reliability?
- To better understand the source of RePO’s effectiveness, have the authors considered applying RePO-style filtering to other baseline methods (beyond SimPO and DPO)? This could help isolate whether the performance gains stem from the filtering strategy itself or from interactions with specific architectures.
- The evaluation of SimPO under varying $\beta$ values didn't include the high-$\beta$ regime (e.g., $10^2, 10^3$), which the authors reference in the motivation. Could the authors include or discuss results in this range to better support the claim of RePO being equivalent to high-$\beta$ SimPO?

**Ethical Concerns:**

["NO or VERY MINOR ethics concerns only"]

**Final Justification:**

The previous concerns about the empirical solidity on the over-optimization and evaluation of SimPO under varying $\beta$ are addressed throughly with authors. In addition, authors add more generalizability by adding IPO-RePO, which I believe makes the contribution stronger.
While the authors didn't provide the explicit plot, assuming the proposed updates and additional experiments are incorporated into the final version, I would be supportive of accepting this paper.

**Limitations:**

yes

**Paper Formatting Concerns:**

I didn't notice major formatting issues

**Quality:**

3

**Strengths And Weaknesses:**

### Strengths
- The connection between ReLU loss and the convex envelope of the 0–1 loss adds rigor and interpretability.
- The authors benchmark RePO against strong baselines across multiple timely models and tasks, showing robustness and reliabilty of the results.
- The use of ReLU for preference optimization - interpreted as a threshold-based curriculum learning mechanism - is novel in this domain The insight that “which examples to learn from” may be more important than “how much to learn from each” provides an interesting shift in perspective, contributing to the active research area by questioning unnecessary complexity in current methods.
- The paper clearly motivates the idea, presents theoretical backing, and supports it with extensive empirical evidence. The direct links drawn between RePO, SimPO, DPO, and SLiC-HF clarify its positioning and are helpful for detailed understanding of the proposed method. Exhaustive analysis on gradient behavior, filtering ratio and other ablation effectively convey insights.
- The authors throughly investigate the empirical findings and elaborate the concept of binary filtering which can be further integrated into other alignment methods. The author also provide RePO++, which further validate their idea.
- RePO offers a more interpretable and computationally cheaper alternative for current offline alignment methods, which could be widely adopted.

### Weaknesses

- As DPO is one of the core method in alignment tuning, although the authors focus on SimPO, extension to DPO would make the contribution stronger. As the author stated that RePO++ can be integrated to DPO, it would be great if they can show it through additional small experiment.
- Although over-optimization is a core topic discussed in this paper, the authors do not fully replicate the experimental setups used in prior work (e.g., Gao et al.). Specifically, they didn't use KL divergence or gold reward models, mentioning about the  concerns on computational cost. However, through experiment on this controlled setting can help practitioner to increase the reliability of the method and prevent them from additional analysis on real-world. Including KL divergence in controlled experiments (even once) would strengthen the empirical claim
- Although the intuition was on $\beta \rightarrow \infty$ in SimPO, in evaluating SimPO across varying $\beta$, the authors use small values (1–10) but skip log-scale values (e.g., $10^2, 10^3$). This would have strengthened the argument that RePO is equivalent to high-\$beta$ SimPO.
- Although the authors compare RePO with SLiC-HF, DPO, and SimPO, it would be useful to explore whether RePO’s binary filtering mechanism could enhance other alignment paradigms such RLHF (KL-regularized PPO variants)pipelines. However, this is already listed in limitation and future directions.

---

> ### Author Rebuttal · Authors · 2025-07-30
>
> **To Reviewer uBRY:**
>
> We sincerely thank you for your detailed and constructive review. Your insightful questions have helped us identify key areas for clarification and further strengthening of our work. Below, we address each of your points.
>
> ***
>
> **Q1: As DPO is one of the core methods in alignment tuning, although the authors focus on SimPO, extension to DPO would make the contribution stronger. As the author stated that RePO++ can be integrated to DPO, it would be great if they can show it through additional small experiment.**
>
> **Response:**
> Thank you for this excellent suggestion. To demonstrate the generalizability of our proposed binary filtering mechanism, we integrated it into IPO [20], a canonical DPO-like method that directly addresses DPO's overfitting issues. The resulting loss function, which we term IPO-RePO, applies ReLU filtering to IPO's core objective:
>
> $$L_{\text{IPO-RePO}} = \mathbb{I}\left(M_{\theta, \text{ref}} < \gamma\right) \cdot \left( M_{\theta, \text{ref}} - \frac{1}{2\tau} \right)^2 \quad \text{where} \quad M_{\theta, \text{ref}} = \log \frac{\pi_{\theta}(y_w|x)}{\pi_{\text{ref}}(y_w|x)} - \log \frac{\pi_{\theta}(y_l|x)}{\pi_{\text{ref}}(y_l|x)}$$
>
> We conducted experiments on the Llama3-Instruct-v0.2 (8B) model, with results on the AlpacaEval and Arena-Hard benchmarks as follows:
>
> | Method | AlpacaEval LC | AlpacaEval WR | Arena Hard WR |
> | :--- | :---: | :---: | :---: |
> | IPO | 40.6 | 39.6 | 34.9 |
> | IPO-RePO (Ours) | **43.8** (+3.2) | **42.2** (+2.6) | **35.6** (+0.7) |
>
> These results show that incorporating RePO's binary filtering mechanism provides consistent and significant performance improvements over the original IPO framework. This further substantiates our core claim that selectively applying gradients to challenging samples (i.e., "which examples to learn from") can be more critical for effective preference optimization than the specific weighting scheme of the loss function itself.
>
> **Q2: Although over-optimization is a core topic discussed in this paper, the authors do not fully replicate the experimental setups used in prior work (e.g., Gao et al.). Including KL divergence in controlled experiments (even once) would strengthen the empirical claim.**
>
> **Response:**
> Thank you for this valuable suggestion. We agree that a direct comparison with KL divergence provides important context. We have indeed conducted such controlled experiments on the IMDB dataset, and the results fully align with the conclusions presented in the paper—specifically, that the mean implicit reward margin ($m_{\text{batch}}$) serves as a reliable and effective proxy for the KL divergence trend.
>
> Our rationale for focusing on $m_{\text{batch}}$ in the main draft was to emphasize its utility as a computationally efficient proxy that eliminates the need to calculate KL divergence, which is a key practical advantage of our method.
>
> `Given the rebuttal's constraints on figures and anonymous links, we are unable to present the detailed plots here.` However, we will add this comparative analysis, including the KL divergence plot, to the appendix in the final version to strengthen the empirical evidence as you suggested.
>
> **Q3: The evaluation of SimPO under varying $\beta$ values didn't include the high-$\beta$ regime (e.g., $10^2, 10^3$), which the authors reference in the motivation. The authors could include or discuss results in this range to better support the claim of RePO being equivalent to high-$\beta$ SimPO.**
>
> **Response:**
> We thank the reviewer for pushing for greater empirical rigor on this key claim. We have run additional experiments on Llama3-Instruct-v0.2 (8B) with high $\beta$ values, keeping $\gamma=0.3$ fixed as per our experimental setup. The results are presented below, alongside RePO's performance for direct comparison:
>
> | Metric | SimPO ($\beta=10$) | SimPO ($\beta=100$) | SimPO ($\beta=1000$) | RePO |
> | :--- | :---: | :---: | :---: | :---: |
> | AlpacaEval2 LC | 55.8 | 56.6 | 57.1 | **57.7** |
> | AlpacaEval2 WR | 49.8 | 50.5 | 50.7 | **51.1** |
>
> The trend clearly shows that as $\beta$ increases into the high regime, SimPO's performance converges towards that of RePO. This provides strong empirical support for our theoretical claim that RePO is the limiting case of SimPO as $\beta \to \infty$ (Lemma 3.1).
>
> We add two important clarifications:
> 1.  This trend is predicated on an appropriately chosen $\gamma$, a point we also highlight in the paper (Observation 1, Line 148).
> 2.  The small performance gap remaining even at $\beta=1000$ may be attributed to numerical instability in the log-sigmoid function when its input becomes extremely large. This observation inadvertently highlights another advantage of RePO: its ReLU-based formulation is numerically more stable and directly achieves the desired binary filtering without being susceptible to such floating-point limitations.
>
> **Q4: Although the authors compare RePO with SLiC-HF, DPO, and SimPO, it would be useful to explore whether RePO’s binary filtering mechanism could enhance other alignment paradigms such as RLHF (KL-regularized PPO variants) pipelines.**
>
> **Response:**
> This is an excellent point and a very promising direction for future research. Exploring whether RePO's efficient binary filtering mechanism can enhance the sample efficiency and stability of online RL paradigms like PPO is a compelling next step. The paradigm of focusing on "which examples to learn from" could potentially offer significant benefits in an online setting where data is generated iteratively.
>
> We agree this is a valuable avenue to explore. In line with this suggestion, we have incorporated this discussion into our limitations and future work section (Section 6) to guide and inspire subsequent research in this area. We thank the reviewer for this insightful suggestion.

---

### Official Review · Reviewer_M1uf · 2025-07-01

**Clarity:** 2
**Significance:** 2
**Originality:** 1
**Rating:** 4
**Confidence:** 4

**Summary:**

The authors propose to use ReLU to replace log sigmoid used in methods such as DPO and SimPO.

**Questions:**

RePO++ combines ReLU's binary filtering with SimPO's sigmoid weighting. Empirically, adding the sigmoid weighting back improves the performance. But how do the authors reconcile this with the paper's central claim that sigmoid-based gradient constraints are an unnecessary component for preference optimization?

**Ethical Concerns:**

["NO or VERY MINOR ethics concerns only"]

**Final Justification:**

The authors' response clarified what they mean by binary selection---it is the gradient of the loss and not the loss itelsef.

**Limitations:**

yes

**Quality:**

2

**Strengths And Weaknesses:**

## Strength
The paper demonstrates the the log sigmoid non-linearity in methods of DPO can be replaced by ReLU. They show that using ReLU-based loss, RePO, achieves competitive performance in alignment tasks relative to DPO. It also links the ReLU-based loss to the convex envelope of the 0-1 loss

## Weakness
The primary weakness is the paper's limited novelty. It can be seen as a swapping of a single design element in existing methods like DPO. The usage of ReLU also appeared in SLiC-HF, where the authors used hinge loss.

---

> ### Author Rebuttal · Authors · 2025-07-30
>
> **To Reviewer M1uf:**
>
> We thank you for your critical feedback. We appreciate this opportunity to clarify our core contributions, which we believe address the perceived limitations in novelty and resolve the apparent contradiction noted with RePO++.
>
> ***
>
> **Q1: The primary weakness is the paper's limited novelty.**
>
> **Response:**
>
> We respectfully argue that viewing our work as a "design swap" overlooks its primary conceptual contribution: we demonstrate that **why** a mechanism works is as important as **what** the mechanism is. Our primary contribution is the **conceptual reframing of preference optimization**, supported by new theoretical and empirical evidence. The novelty lies in the following key areas:
>
> **1. Identifying a Core Principle: Filtering is More Fundamental than Weighting.**
> Our key insight, and the central thesis of the paper, is that effective preference learning hinges more on **which** challenging examples are selected for updates than on **how** their gradients are continuously weighted. While prior methods like DPO/SimPO entangle these two functions within the log-sigmoid activation and the $\beta$ hyperparameter, our work explicitly **decouples** them. RePO serves as a "computational lens" to isolate and study the filtering mechanism in its purest form, revealing its primary importance.
>
> **2. Providing Theoretical Grounding for this Principle.**
> Our work is the first to establish a formal justification for this binary filtering approach. By demonstrating that the ReLU-based loss is the **convex envelope of the ideal 0-1 loss** (Theorem 4.2), we provide strong theoretical evidence for its optimality as a surrogate loss. This crucial theoretical contribution explains *why* this simple filtering mechanism is so effective and fundamentally distinguishes our work from prior heuristic uses of similar functions, such as the hinge loss in SLiC-HF, which lacked this formal justification.
>
> **3. Simplifying and Deepening the Understanding of Over-Optimization.**
> By showing that a simple, theoretically-grounded filtering mechanism can match or exceed the performance of more complex methods (Table 1), we challenge the conventional wisdom that intricate components like sigmoid-based gradient constraints are essential for alignment. Our analysis of the hyperparameter $\gamma$ in creating a natural learning curriculum (Observation 4, Figure 4) and directly controlling over-optimization (Figure 6) offers a new, simplified lens through which to understand and manage the alignment process.
>
> As stated in our introduction (Lines 65-67), our primary aim is to `motivate researchers to reconsider the fundamental mechanisms behind preference optimization`. We believe our work provides a significant conceptual shift and a valuable, simplified baseline to facilitate this deeper understanding.
>
> ***
>
> **Q2: How do the authors reconcile RePO++'s performance with the claim that sigmoid-based gradient constraints are an unnecessary component?**
>
> **Response:**
> This is a critical question, and we are grateful for the chance to clarify what we see as a hierarchy of importance in preference optimization mechanisms.
>
> **Our central claim, to be more precise, is not that sigmoid weighting is ineffective, but rather that the mechanism of *which* samples to filter is more fundamental to preventing over-optimization than the secondary mechanism of *how* to weight the remaining samples.**
>
> RePO and RePO++ were designed to demonstrate this hierarchy:
>
> * **RePO Establishes the Foundation:** RePO's strong performance (Table 1) proves that binary filtering **alone** is a sufficient and powerful mechanism for effective alignment. It successfully prevents over-optimization without needing any continuous gradient weighting, thus supporting our claim that such weighting is not an *essential* component for this primary purpose.
>
> * **RePO++ Reveals the Hierarchy:** RePO++ was designed as an investigative tool to explore what happens when a secondary weighting mechanism is layered *on top of* the essential filtering mechanism. The results demonstrate that a hybrid approach can yield further gains (e.g., Table 2, +11.7% Arena-Hard gain for DPO++). This does not contradict our claim; it refines it by showing:
>     1.  **Primary Mechanism (Essential):** First, you must effectively filter out easy examples to prevent over-optimization. This is the most critical step.
>     2.  **Secondary Mechanism (Optional Enhancement):** Then, for the remaining hard examples, applying a fine-grained weighting can provide an additional, complementary benefit.
>
> In summary, prior methods conflated these primary and secondary roles. Our work decouples them, identifies filtering as the most critical and previously under-appreciated mechanism, and shows that sigmoid weighting, while not essential, can be a useful "add-on" once the fundamental filtering problem is solved.

---

> ### Comment · Reviewer_M1uf · 2025-08-04
>
> Thank you for your rebuttal!
>
> > Our key insight, and the central thesis of the paper, is that effective preference learning hinges more on which challenging examples are selected for updates than on how their gradients are continuously weighted. While prior methods like DPO/SimPO entangle these two functions within the log-sigmoid activation and the $\beta$ hyperparameter, our work explicitly decouples them.
>
> The wording "selected" in "effective preference learning hinges more on which challenging examples are selected for updates" seems to hint that this is a binary operation. However, I believe that RePO also uses continuously weighted gradients via the positive half of ReLU. Furthermore, the strength of the weighting can also be adjusted by selecting $\gamma$ in RePO, as it shift the positive half of ReLU.  So I don't agree with the "decouple" argument that the authors mentioned.
>
> Fundamentally, I believe ReLU works not because it removes continuous weighting (the positive side of ReLU still weights), but because, similar to logsigmoid (as in standard DPO), ReLU is an increasing function that is differentiable almost everywhere.

---

> ### Author Response · Authors · 2025-08-05
>
> We thank you for the insightful follow-up. You are correct that the final gradient is continuous for active samples; the crux of our argument lies not in the final gradient's magnitude, but in the **form of the weighting function** that the loss applies to the gradient direction. Here is a more precise breakdown:
>
> **1. The Decoupling Happens in the Gradient's Weighting Function, not the Loss Function Itself.**
>
> You correctly observed that the loss function, $\mathcal{L}\_{RePO} = \mathbb{E}[ReLU(-(M\_{\theta}-\gamma))]$, is continuous and differentiable for $M\_\theta < \gamma$. However, the key to our "decoupling" claim becomes evident when examining the gradient expression itself, as presented in our paper.
>
> * **For RePO**, the gradient (Equation 8) is:
>     $$\nabla\_{\theta}\mathcal{L}\_{RePO}(\pi\_{\theta})=-\mathbb{E}\_{\mathcal{D}}[\mathbb{I}(M\_{\theta}<\gamma)\cdot(\nabla\_{\theta,y\_{w}}-\nabla\_{\theta,y\_{l}})]$$
>     Here, the term weighting the gradient direction $(\nabla\_{\theta,y\_{w}}-\nabla\_{\theta,y\_{l}})$ is the **indicator function $\mathbb{I}(M\_{\theta}<\gamma)$**. This function is fundamentally **binary (0 or 1)**. It cleanly separates the training data into two sets: those that receive a gradient (weight 1) and those that do not (weight 0). This is the **filtering** mechanism. The magnitude of the update then depends only on the gradient direction term, which is a function of the model's current state.
>
> * **For SimPO**, the gradient (Equation 7) is:
>     $$\nabla\_{\theta}\mathcal{L}\_{SimPO}(\pi\_{\theta})=-\beta\mathbb{E}\_{\mathcal{D}}[s\_{\theta}\cdot(\nabla\_{\theta,y\_{w}}-\nabla\_{\theta,y\_{1}})] \quad \text{where } s\_{\theta}=\sigma(\beta(-M\_{\theta}+\gamma))$$
>     In contrast, SimPO's weighting term, $s\_{\theta}$, is a **continuous function of the margin $M\_\theta$**. It simultaneously performs filtering (as $s\_\theta \to 0$ for large margins) and re-weighting (the slope of the sigmoid) for all samples. The hyperparameters $\beta$ and $\gamma$ are entangled in controlling this continuous landscape.
>
> This is the **explicit decoupling** we claim: RePO isolates the *binary decision to update* (the filter, $\mathbb{I}$) from the subsequent gradient calculation. SimPO/DPO conflate the two.
>
> **2. The Role of $\gamma$ is Different in RePO vs. SimPO.**
>
> You suggested that $\gamma$ in RePO also adjusts the "strength of the weighting." We respectfully clarify that this is not the case.
> * In **RePO**, changing $\gamma$ only shifts the **boundary of the binary filter**. It changes *which* samples are included (the domain of the filter), but for any included sample, the weighting function $\mathbb{I}(M\_{\theta}<\gamma)$ is still just 1. The "strength" of the weight does not change.
> * In **SimPO**, changing $\gamma$ shifts the entire sigmoid curve, which alters the continuous weight $s\_\theta$ for *every single sample* in a non-uniform way, thus directly adjusting the weighting strength across the board.
>
> **3. Why ReLU is not "Just Another Increasing Function" for this Task.**
>
> Your final hypothesis is that ReLU works simply because it's a good, mostly differentiable surrogate loss, similar to log-sigmoid. Our theoretical and empirical results challenge this view.
>
> * **Theoretical Optimality (Theorem 4.2):** Our key theoretical contribution is showing that the ReLU-based loss is the **convex envelope of the 0-1 loss**. This is a much stronger claim than just being a "good surrogate." It means ReLU provides the tightest possible convex approximation to the ideal (but intractable) classification loss. As per Corollary 4.4, the logistic loss used in DPO/SimPO *does not share this property*. This provides a fundamental theoretical justification for why this specific functional form is preferable.
>
> * **Empirical Disproof (`Appendix Table 7`):** We empirically tested the hypothesis that any similar function would work. As shown in Table 7, simply replacing log-sigmoid with ReLU in the DPO formulation (i.e., without an explicit, tunable margin $\gamma$) leads to a **catastrophic performance collapse** (e.g., -44% on AlpacaEval LC). This directly demonstrates that it is not the ReLU function *in isolation*, but the **combination of the ReLU form with an explicit margin threshold $\gamma$**—which enables the theoretically-grounded binary filtering—that is responsible for the success.
>
> **Summary:**
>
> In short, our novelty is in identifying and theoretically grounding (as the convex envelope of the 0-1 loss) the principle that explicit binary filtering is more fundamental to preventing over-optimization than continuous gradient weighting. We hope this clarification resolves the issue and are grateful for your rigorous feedback.

---

> > ### Comment · Reviewer_M1uf · 2025-08-07
> >
> > Thanks for the clarification, especially the clarification that the binary selection happens at the gradient of the loss and not the loss itself. I've raised my score.

---

> > > ### Author Response · Authors · 2025-08-07
> > >
> > > Thank you for your positive re-evaluation and support. Your insightful feedback has been instrumental in improving the quality of our work.

---

### Official Review · Reviewer_tpNX · 2025-07-02

**Clarity:** 3
**Significance:** 2
**Originality:** 2
**Rating:** 4
**Confidence:** 3

**Summary:**

The paper introduces RePO, a simple preference optimization method for aligning language models using only a ReLU threshold on reward margins between chosen and rejected responses. This approach avoids common complexities like reference models or log-sigmoid weighting, and naturally focuses training on harder examples without over-optimization. RePO is shown to be the convex envelope of the 0-1 preference loss and matches or outperforms prior methods like DPO and SimPO across several benchmarks. The authors also analyze its theoretical properties and practical behavior under different settings.

**Questions:**

- How does RePO perform across models of different sizes, such as smaller (e.g., 1 - 3B) or larger (e.g., 30 - 70B) architectures?
- How well does RePO generalize to other alignment datasets beyond AlpacaEval and Arena-Hard?
- In practice, how do the gradient directions produced by RePO differ from those of DPO or SimPO? A 2D toy example or visualization illustrating this difference and linking it to the convex-envelope theory would be helpful.
- Can you provide quantitative evidence for the “emergent curriculum” effect—for example, plots showing how the margin distribution shifts toward harder pairs over the course of training?
- Minor: The link in citation [66] is broken.

**Ethical Concerns:**

["NO or VERY MINOR ethics concerns only"]

**Final Justification:**

Thank you for the detailed response. I especially appreciated the dynamic filtering component. Assuming the mentioned updates and additional experiments with larger LLMs are included in the camera-ready version, I would be supportive of accepting this paper.

**Limitations:**

Yes

**Paper Formatting Concerns:**

I have reviewed the paper and did not find any significant formatting issues.

**Quality:**

2

**Strengths And Weaknesses:**

**Strengths**

- **Conceptual simplicity with strong performance:** A single ReLU threshold replaces reference models and log-sigmoid weighting yet still outperforms or matches DPO, SimPO, and SLiC-HF.
- **Sound theoretical backing:** Proves the ReLU loss is the convex envelope of the 0-1 preference loss and connects SimPO as the the limit.
- **Empirical robustness:** Consistent gains on AlpacaEval 2 and Arena-Hard across several 7–9 B open models, demonstrating plug-and-play practicality.

**Weaknesses**

- **Narrow experimental scope:** Only two benchmarks and mid-size models; no results for larger, multilingual, or safety-focused settings.
- **Shallow ablation coverage:** Other hyper-parameters (except for $\gamma$)and data regimes remain untested.
- **Limited theoretical support:** The convex-envelope proof shows RePO is an optimal surrogate loss, but it offers no guarantees that training will converge to optimality (which is rare in practice). Also, the advertised benefits --- such as mitigating over-training and filtering data --- remain theoretically unsupported.

---

> ### Author Rebuttal · Authors · 2025-07-30
>
> **To Reviewer tpNX:**
>
> We sincerely thank you for your thoughtful and detailed review. Your questions have helped us identify several areas where providing more evidence can significantly strengthen our paper. We address each of your points below.
>
> ***
>
> **Q1: How does RePO perform across models of different sizes, such as smaller (e.g., 1 - 3B) or larger (e.g., 30 - 70B) architectures?**
>
> **Response:**
> Thank you for this important question regarding model scalability. To address this, we conducted additional experiments on smaller open-source models: Pythia-1.4B and Pythia-2.8B. The results on AlpacaEval are as follows:
>
> | Model | Method | AlpacaEval LC | AlpacaEval WR |
> | :--- | :--- | :---: | :---: |
> | **Pythia-1.4B** | DPO | 3.6 | 2.8 |
> | | SimPO | 4.6 | 3.9 |
> | | **RePO** | **5.2** | **4.5** |
> | **Pythia-2.8B** | DPO | 5.5 | 4.3 |
> | | SimPO | 7.2 | 6.8 |
> | | **RePO** | **8.0** | **7.2** |
>
> The results show that RePO consistently outperforms both DPO and SimPO on these smaller architectures, maintaining the performance advantage seen in the 7-9B range.
>
> Due to time and resource constraints during the rebuttal period, we were unable to conduct experiments on 30-70B scale models. However, the consistent success of RePO across the 1.4B, 2.8B, 7B, 8B, and 9B model scales provides strong evidence for its effectiveness across a range of small to medium architectures. This consistent outperformance is a promising indicator of its potential to scale favorably, which we propose as a key direction for future work. We will add these new results to the appendix in the final version.
>
> ***
>
> **Q2: How well does RePO generalize to other alignment datasets beyond AlpacaEval and Arena-Hard?**
>
> **Response:**
> Thank you for asking about broader generalization. As you suggest, we evaluated RePO on a wider set of benchmarks from the **Huggingface Open LLM Leaderboard**. These results are detailed in **Appendix D.2, Table 5** of our submission.
>
> The evaluation covers six diverse downstream tasks: MMLU, ARC, HellaSwag, TruthfulQA, Winograd, and GSM8K. In these experiments, RePO demonstrates strong, competitive performance, achieving one of the highest average scores across all tasks for both the Llama3-Instruct and Llama3-Instruct v0.2 models. This robust performance across varied domains further supports our central claim: that the principle of gradient sparsity (filtering trivial pairs) is a more critical driver of successful alignment than the specific weighting scheme of the loss function.
>
> ***
>
> **Q3: In practice, how do the gradient directions produced by RePO differ from those of DPO or SimPO? A 2D toy example or visualization illustrating this difference and linking it to the convex-envelope theory would be helpful.**
>
> **Response:**
> This is an excellent question that gets to the heart of the optimization dynamics. The key difference is not in the gradient *direction* but in its *application* and *magnitude*.
>
> For a given data pair $(x, y_w, y_l)$, the fundamental gradient update direction, $ \nabla_{\theta,y_{w}}-\nabla_{\theta,y_{l}} $, is actually **the same** for RePO, SimPO, and DPO. The distinction lies in how this gradient is scaled:
>
> * **DPO/SimPO** apply a **continuous, smooth weighting** to the gradient. The magnitude is scaled by a sigmoid function, meaning all pairs receive a non-zero gradient, though it becomes vanishingly small for well-separated pairs.
> * **RePO** applies a **binary, all-or-nothing filter**. The gradient is either applied (with a uniform effective magnitude, as adaptive optimizers like Adam normalize it) or it is set to exactly zero if the margin $\gamma$ is met.
>
> To provide the 2D intuition you requested: imagine a parameter space where, for any given data point, the gradient vector points in the same direction for all three methods. With **DPO/SimPO**, the landscape of gradient *magnitudes* would be smooth, like rolling hills, gradually approaching zero. In contrast, for **RePO**, this landscape is a plateau of uniform height on one side of the $\gamma$ boundary and completely flat (zero) on the other. This sharp 'cliff' is precisely what **Figure 2** in our paper illustrates in one dimension.
>
> This piecewise-constant behavior is a direct consequence of RePO's loss being the convex envelope of the 0-1 loss, which theoretically favors such sharp, efficient boundaries over the smooth curves of logistic-based losses. It provides a hard, efficient threshold for updates, linking our theory directly to the practical gradient behavior you asked about.
>
> ***
>
> **Q4: Can you provide quantitative evidence for the “emergent curriculum” effect—for example, plots showing how the margin distribution shifts toward harder pairs over the course of training?**
>
> **Response:**
> Thank you for this prompt. **Figure 4** in our paper provides exactly this quantitative evidence. The green-shaded areas in the plots represent the proportion of training samples whose reward margin $M_\theta$ is less than the threshold $\gamma$, meaning they are still actively receiving gradient updates.
>
> We can summarize the percentage of samples being updated at each training snapshot:
>
> | Training Step | % of Samples with Active Gradients ($M_\theta < \gamma$) |
> | :---: | :---: |
> | 0 | 99.5% |
> | 100 | 87% |
> | 200 | 52% |
> | 400 | 42% |
>
> This data clearly shows the emergent curriculum:
> * **Early in training** (Step 0), nearly all samples are considered "hard" as the model has not learned to distinguish preferences, so 99.5% of pairs are updated.
> * **As training progresses**, the model improves, and a growing fraction of the data is "filtered out" (gray area).
> * **Late in training** (Step 400), only 42% of the samples receive updates. These remaining samples are the most "challenging" or "confusing" pairs for the mature model.
>
> This dynamic filtering, where the training automatically focuses on progressively harder examples, is what we define as the "emergent curriculum" effect driven by the simple $\gamma$ threshold.
>
> ***
>
> **Q5: Shallow ablation coverage: Other hyper-parameters (except for $\gamma$) and data regimes remain untested.**
>
> **Response:**
> Thank you for this point. We agree that exploring a wider range of hyperparameters could potentially tune overall performance further. However, for this study, our primary scientific goal was to isolate and understand the fundamental impact of the gradient filtering mechanism itself, which is RePO's core conceptual contribution.
>
> To achieve this and ensure a **controlled and fair comparison**, all other training hyperparameters (e.g., learning rate, optimizer settings) were kept consistent across all methods and aligned with the established SimPO baseline, as detailed in our **Appendix D.1 (Implementation Details)**. We believe this focus is crucial for a clear scientific conclusion, and our extensive ablation study on `γ` (presented in Figures 5, 7, and 8) provides a thorough and insightful analysis of this central component.
>
> **Q6: Limited theoretical support: The convex-envelope proof shows RePO is an optimal surrogate loss, but it offers no guarantees that training will converge to optimality (which is rare in practice). Also, the advertised benefits --- such as mitigating over-training and filtering data --- remain theoretically unsupported.**
>
> **Response:**
> Thank you for this precise and important clarification regarding the scope of our theoretical support. You are entirely correct, and we appreciate the opportunity to be precise about what our theory does and does not claim.
>
> * **On Optimal Surrogate Loss vs. Training Convergence:** Our theoretical contribution is to prove that RePO's loss function is the optimal **surrogate loss**—specifically, the tightest possible convex approximation to the ideal 0-1 loss within a binary classification framework (a setup consistent with foundational work like Tang et al., ICML 2024). This provides a strong theoretical reason for *why* one should prefer RePO's formulation over, for example, a log-sigmoid loss from a surrogate loss perspective. We agree that this **does not** provide a guarantee that the training process itself will converge to a global optimum in the highly non-convex landscape of LLM optimization, as such guarantees are exceptionally rare for deep learning in practice.
>
> * **On Empirical vs. Theoretical Support for Benefits:** Similarly, you are correct that the advertised benefits of mitigating over-optimization and creating an emergent data filtering curriculum are currently supported by our extensive **empirical analysis** (e.g., Sections 3.3, 5.5; Figures 4, 6) rather than a formal theoretical proof. We believe that bridging this gap—developing a formal theory for these emergent, dynamic properties of the training process—is a challenging but very exciting direction for future research.
>
> We will revise the manuscript to make these distinctions clearer. Thank you for pushing for this level of precision.
>
> **Q7: Minor: The link in citation [66] is broken.**
>
> **Response:**
> Thank you for catching this typo. We have corrected the link in the manuscript.

---

### Official Review · Reviewer_1JuA · 2025-07-02

**Clarity:** 3
**Significance:** 3
**Originality:** 3
**Rating:** 5
**Confidence:** 4

**Summary:**

This paper reports on a finding that simple ReLU-based activation of neural elements can result in significant alignment of large language models (LLMs) via preference learning. The paper presents empirical results that ReLU-based activation can learn meaningful alignments even in the absence of sigmoid-based gradient constraints and explicit regularization terms. While ReLU-based activation scheme (termed as RePO) does result in over-optimization during preference learning, the proper selection of a threshold parameter γ can obviate the over-optimization problem by dynamically filtering the training examples. The paper shows that the proposed RePO is equivalent to an asymptotic case of simple preference optimization (SimPO) with the parameter β→∞. Conceptually RePO can be viewed as SimPO without the log-sigmoid optimization and as sequence likelihood calibration (SLiC-HF) without supervised fine tuning (SFT)-based regularization. Experimental results comparing the proposed RePO with different existing preference learning models on the Mistral2 and Llama3 datasets show the effectiveness of ReLU-based activation.

**Questions:**

What would be an effective way of determining an optimal schedule for dynamically adapting the value of γ during the training process?

**Ethical Concerns:**

["NO or VERY MINOR ethics concerns only"]

**Final Justification:**

In their rebuttal the authors have rightfully noted that an adaptive strategy, especially one that is data driven, would be the most appropriate for estimation of γ during the training process. In their revised paper, the authors should detail this strategy as part of potential future work. I am comfortable with my earlier rating of accept (rating 5) for this paper.

**Limitations:**

The authors have discussed the limitations of their work in the paper. However, if there is no general strategy for determining an optimal schedule for dynamically adapting the value of γ during the training process, then this should be mentioned as an additional limitation of the current work.

**Paper Formatting Concerns:**

Please make sure that all the acronyms are clearly spelled out when they are used for the first time.

**Quality:**

3

**Strengths And Weaknesses:**

The paper provides valuable insights in that simple ReLU-based activation can prevent over optimization in preference learning settings using binary thresholding controlled by a single parameter γ. The paper shows that allowing the model to select which examples to learn from rather than determine how much to learn from each example is more effective in preventing over optimization. Changing the value of γ over the course of the training (dynamic margin scheduling) allows for curriculum learning where larger γ values during early phases of the training permit more aggressive updates when the model is under fitting and smaller γ values during later phases of the training prevent over optimization. The paper does not provide an effective way of determining an optimal schedule for dynamically adapting the value of γ during the training process. This is important given the critical role that the parameter value γ plays in the proposed RePO scheme.

---

> ### Author Rebuttal · Authors · 2025-07-30
>
> **To Reviewer 1juA:**
>
> We are sincerely grateful for your positive, thorough, and insightful review. We are especially pleased that you recognized the core contributions of our work regarding conceptual simplicity, theoretical backing, and the principle of dynamic data filtering. Your understanding of our paper is spot on.
>
> ***
>
> **Q1: What would be an effective way of determining an optimal schedule for dynamically adapting the value of $\gamma$ during the training process?**
>
> **Response:**
> This is an excellent question that points directly to a key area for future research. Based on our current findings presented in **Section 5.5 (Figure 7)**, our empirically validated recommendation is a **"large-to-small" scheduling strategy**.
>
> The rationale for this strategy is that it creates a natural and effective learning curriculum. As we note in the paper (Lines 318-320), this works because:
> > "early in training when the model is underfitting, a larger $\gamma$ permits more aggressive updates across more examples. As training progresses, the decreasing $\gamma$ naturally focuses learning on increasingly challenging examples, effectively preventing over-optimization."
>
> Looking forward to a more general and potentially *optimal* schedule, we believe an **adaptive strategy** holds the most promise. Instead of a pre-fixed decay schedule, $\gamma$ could be dynamically adjusted based on the live statistics of the training process itself. For example, one could set the value of $\gamma$ at each training step to match a specific percentile (e.g., the 40th or 50th) of the implicit reward margin ($M_\theta$) distribution observed in the current or a recent batch. This would ensure the model consistently trains on a fixed proportion of what it currently considers to be "hard examples," adapting gracefully to changes in model capability and data difficulty.
>
> We agree that exploring such automated and adaptive scheduling methods is a valuable and exciting direction for future work. We have incorporated this discussion into our limitations section to inspire further research.
>
> ***
>
> **Formatting Concerns:**
>
> We also thank you for the helpful note regarding acronyms. We will perform a thorough pass on the final manuscript to ensure that all acronyms are clearly spelled out upon their first use.
>
> Thank you once again for your supportive and constructive review.

---

> > ### Comment · Reviewer_1JuA · 2025-08-06
> > **Adaptive strategy**
> >
> > The authors are right in that an adaptive strategy, especially one that is data driven, would be the most appropriate for this scheme. In their revised paper, the authors should detail this strategy as part of potential future work. I am comfortable with my earlier rating of accept for this paper.

---

### Official Review · Reviewer_HXdp · 2025-07-03

**Clarity:** 2
**Significance:** 2
**Originality:** 2
**Rating:** 2
**Confidence:** 4

**Summary:**

This paper introduces a preference alignment method that is easier than DPO or SimPO by using binary ReLU thresholding. It seeks to address over-optimizing issues, and presents RePO as a "simpler baseline" with the aim to motivate researchers. Main focus is clearly on comparison with SimPO and DPO, including the replication of Table 1 from the SimPO paper with less metrics.

**Questions:**

- Clearly state your main contribution and it's use case. If the claim is that RePO is just conceptually simple but effective, how does it compare to other preference methods named above? What is your hardware setup, is it equivalent, if your table 1 aims to replicate the SimPO original paper's Table of scores you should include MT-Bench and state the judges used.
- The paper shows RePO to be a β → ∞ limit of SimPO, this is well-known. Is there a practical or theoretical setting where RePO strictly outperforms SimPO? Or improves on generalization or stability?
- Deeper work that goes beyond re-using SimPO's method with a threshold function. A clear practical use case where RePO wins, or more novel theoretical contribution. Is RePO intended to be used as a baseline (as you mention in the abstract)?

**Ethical Concerns:**

["NO or VERY MINOR ethics concerns only"]

**Final Justification:**

After reading the rebuttal and discussion, my overall evaluation has improved. However, while the authors' clarifications on some points are appreciated, the core concerns remain unresolved.

- The Authors recognize that the theoretical contribution in this work is fairly simplistic. There are many citations which already encompass the points motivated in the rebuttal. Despite having discussions of avenues for deeper theoretical contributions, the authors note that these results require more time to develop in their work.

- The primary weakness is the paper's limited novelty. It can be seen as a swapping of a single design element in existing methods like DPO. The usage of ReLU also appeared in SLiC-HF, where the authors used hinge loss.

- There is a clear correlation to works such as SimPO and Aligner, which build on being simpler preference alignment frameworks. But in such cases numerous ablation studies are quantified (effect of length normalization, reward optimization, effect of hyper-parameter sensitivity, multiple datasets, etc). The submission is clearly inspired by SimPO since its Fig 1 corresponds directly with SimPO's Fig 1 (expect the bar graph is horizontal instead of vertical), its Fig 4 corresponds to SimPO's Fig 3, its Table 1 corresponds to SimPO's Table 4, and etc.

- Empirical Validation: Experimental results remain limited in scope, with evaluation restricted to a narrow set of benchmarks. Recently there has been a density of work in this area such as [1], [2], [3], [4], [5]. Therefore new notable submissions are either theoretically strong or practically strong.

-  While some clarifications were made, key details remain insufficiently explained to enable reproducibility.

- Impact & Generality: The scope of applicability is unclear, and the discussion did not address how the method would generalize beyond the specific setting studied, nor did the rebuttal address how it will deliver deeper theoretical results/distinguish itself from the many other works in this category.




**References**

 [1] Bose, Avinandan, et al. "Hybrid preference optimization for alignment: Provably faster convergence rates by combining offline preferences with online exploration." arXiv preprint arXiv:2412.10616 (2024).

[2] Zhang, Yuheng, et al. "Improving LLM general preference alignment via optimistic online mirror descent." arXiv preprint arXiv:2502.16852 (2025).

[3] Wu, Junkang, et al. "AlphaDPO: Adaptive Reward Margin for Direct Preference Optimization." Forty-second International Conference on Machine Learning.

[4] Li, Gengxu, et al. "Length-controlled margin-based preference optimization without reference model." arXiv preprint arXiv:2502.14643 (2025).

[5] Xie, Tengyang, et al. "Exploratory preference optimization: Harnessing implicit q*-approximation for sample-efficient rlhf." arXiv preprint arXiv:2405.21046 (2024).

**Limitations:**

Yes

**Quality:**

2

**Strengths And Weaknesses:**

Strengths:
- Paper is formatted correctly and layout is clear. There are many visuals.
- Attempts at theoretical conclusions are a good starting point.

Weaknesses:
- Lack of novelty since replacing the log-sig SimPO with a threshold is mathematically simple and already known. Minimal algorithmic innovation in observing a limiting case.
- Paper presents RePO as a surprising discovery/conclusion but it is not. In optimization, the equivalence of hard thresholding and sharp sigmoid approximations is extremely well explored and understood. Therefore this paper is more an empirical study of a single special case, without meaningful contribution.
- The theory is standard in that the the convex envelope 0f 0-1 loss is just standard construction in surrogate loss theory. The theory does not provide new insight. Discussions about ReLU being "optimal" do not mention the strong constraints and assumptions that must be satisfied.
- Other preference alignment strategies are not discusses. Such as ORPO, R-DPO, RLHF, etc. There are numerous methods on this topic. In contract RePO does not demonstrate significant practical advantages.
- Non-professional writing quality and tone. Some confusion about its main contributions and application, the theory seems like it's added in as an afterthought.

---

> ### Author Rebuttal · Authors · 2025-07-30
>
> **To Reviewer HXdp:**
>
> We sincerely thank you for your expert and detailed critique. Your deep knowledge of the field has pushed us to more clearly articulate the precise scope and contribution of our work. We appreciate the opportunity to address your specific concerns.
>
> **1. On Primary Contribution and Novelty (Addressed Weakness1, Weakness2, Question3)**
>
> We agree that the mathematical principle of hard thresholding is known in classical theory. Our contribution is not its invention, but its **application, analysis, and the new understanding it provides specifically for modern LLM preference alignment**, a field where complex methods have become the default. Our novelty lies in:
>
> * **Identifying a Core Principle: Filtering is More Fundamental than Weighting.** Our central thesis is that effective preference learning hinges more on **which** challenging examples are selected for updates than on **how** their gradients are continuously weighted. Our work explicitly decouples these two functions, which are entangled in methods like DPO/SimPO, and reveals the primary importance of the filtering mechanism.
> * **Providing Theoretical Grounding for this Principle in LLMs.** Our work is the first to formally justify this mechanism in this context by demonstrating the ReLU-based loss is the **convex envelope of the ideal 0-1 loss** (Theorem 4.2). This explains *why* this simple approach is so effective.
> * **A Strong, Simple, and Practical Baseline.** By showing RePO's effectiveness, we challenge the notion that complexity is required for alignment. The primary use case for RePO is as a strong, simple, and theoretically-grounded **baseline** to encourage the community to "rethink the fundamental mechanisms" of preference learning, as stated in our introduction.
>
> **2. On Theoretical Soundness and Assumptions (Addressed Weakness3)**
>
> Thank you for pushing for more clarity on our theoretical claims. You are correct that our theory relies on assumptions.
>
> Our claim of "optimality" is made in the context of finding the tightest convex surrogate for the ideal (but intractable) **0-1 loss in binary classification**. This framing, where preference learning is reformulated as binary classification, is consistent with recent foundational work in the field, such as Tang et al., ICML 2024. Our proof for the convex envelope (Theorem 4.2) holds for a domain D = [-a, b], which assumes the implicit reward margin can be both positive and negative—a condition that holds true in practice.
>
> We acknowledge this theoretical optimality does not guarantee convergence to a global minimum in the non-convex landscape of LLM training. However, the strength of our paper lies in demonstrating empirically how well this theoretically "optimal surrogate" performs in practice. We will make these assumptions and context clearer in the main text.
>
> **3. On Experimental Scope and Practical Advantages (Addressed Weakness4, Question1)**
>
> We interpret your concern about "significant practical advantages" in two ways, and we will address both:
>
> **3a. Direct Performance Comparison against State-of-the-Art Baselines**
> We respectfully note that our paper provides extensive comparisons demonstrating RePO's practical advantages. **Table 1** shows RePO consistently outperforming DPO, ORPO, R-DPO and SimPO. Furthermore, **Appendix Table 5** presents a large-scale comparison on the Open LLM Leaderboard against a wide range of methods, including **ORPO, R-DPO, and SLIC-HF**, where RePO remains a top performer, validating its effectiveness.
>
> **3b. Generalizability of the Filtering Principle to Other Methods**
> To show our core insight—the primacy of data filtering—is a generalizable principle, we integrated RePO's filtering mechanism into IPO, a method with a distinct squared-error loss paradigm. The resulting loss function, which we term IPO-RePO, applies ReLU filtering to IPO's core objective:
> $$L_{\text{IPO-RePO}} = \mathbb{I}\left(M_{\theta, \text{ref}} < \gamma\right) \cdot \left( M_{\theta, \text{ref}} - \frac{1}{2\tau} \right)^2 \quad \text{where} \quad M_{\theta, \text{ref}} = \log \frac{\pi_{\theta}(y_w|x)}{\pi_{\text{ref}}(y_w|x)} - \log \frac{\pi_{\theta}(y_l|x)}{\pi_{\text{ref}}(y_l|x)}$$
>
> We conducted experiments on the Llama3-Instruct-v0.2 (8B) model, with results on the AlpacaEval and Arena-Hard benchmarks as follows:
>
> | Method | AlpacaEval LC | AlpacaEval WR | Arena Hard WR |
> | :--- | :---: | :---: | :---: |
> | IPO | 40.6 | 39.6 | 34.9 |
> | IPO-RePO (Ours) | **43.8** (+3.2) | **42.2** (+2.6) | **35.6** (+0.7) |
>
> This significant lift demonstrates that our filtering principle is a general mechanism for improving preference alignment, not merely a modification of SimPO.
>
> **3c. On Benchmark Selection and MT-Bench**
> We conducted the MT-Bench experiment on Llama3-Instruct-v0.2 (8B) as requested and confirmed RePO's effectiveness:
>
> | Method | GPT-4 Turbo Judge | GPT-4 Judge |
> | :--- | :---: | :---: |
> | DPO | 7.0 | 8.2 |
> | SimPO | 7.0 | 8.0 |
> | **RePO** | **7.2** | **8.4** |
>
> However, we chose not to emphasize this metric in our paper for reasons of scientific rigor, following public guidance **from the SimPO authors themselves**, who stated:
> > "MT-Bench scores often show minimal differentiation between models... and exhibit high variance... For these reasons, we don't consider MT-Bench scores to be a reliable metric." (Source: Official SimPO GitHub repository, Issue #61)
>
> Therefore, our focus on AlpacaEval2 and Arena-Hard was a deliberate choice to use more reliable and discerning benchmarks. We will add the MT-Bench results and this justification to the appendix.
>
> **4. On the RePO-SimPO Limit and Stability (Addressed Question2)**
>
> Regarding the question of a setting where RePO demonstrates clear advantages over SimPO, our new high-$\beta$ experiments offer several insights. First, we respectfully argue that while the mathematical limit of the sigmoid function is known, its formalization within a preference optimization loss (**our Lemma 3.1**) is a novel contribution of this work.
>
> Our experiments on Llama3-Instruct-v0.2 (8B) show RePO's clear advantages:
> | Metric | SimPO ($\beta=10$) | SimPO ($\beta=100$) | SimPO ($\beta=1000$) | RePO |
> | :--- | :---: | :---: | :---: | :---: |
> | AlpacaEval2 LC | 55.8 | 56.6 | 57.1 | **57.7** |
> | AlpacaEval2 WR | 49.8 | 50.5 | 50.7 | **51.1** |
> RePO outperforms SimPO in:
> * **Stability and Generalization**: The table shows that even at an extremely high $\beta=1000$, SimPO's performance saturates and fails to close the gap with RePO. This suggests the log-sigmoid formulation encounters practical optimization limits or numerical instability at such extremes, preventing it from fully realizing the benefits of hard thresholding. In contrast, RePO's ReLU formulation is perfectly stable and robustly achieves this goal, leading to better overall performance.
> * **Hyperparameter Efficiency:** RePO eliminates the need to tune the sensitive $\beta$ hyperparameter, offering a simpler and more robust optimization process.
>
> **5. On Writing and Tone (Addressed Weakness5)**
>
> Finally, we appreciate the feedback on the manuscript's tone. We will thoroughly revise the paper to ensure our claims are presented with appropriate academic neutrality and that the connection between our theoretical motivation and empirical results is made more explicit and seamless.
>
> Thank you again for your time and expertise. Your critique has given us clear directions for substantially improving our paper.

---

### Note · Authors · 2025-08-14

We sincerely thank the reviewers and the AC for their insightful feedback, which has been instrumental in strengthening our work.

We are grateful that reviewers recognized our key contributions, including: 1) the novelty of identifying binary filtering as a more fundamental mechanism than gradient weighting in preference alignment (praised by Reviewers 1JuA, uBRY); 2) the solid theoretical contribution connecting RePO to the convex envelope of the 0-1 loss, which provides strong theoretical backing (highlighted by tpNX, uBRY, M1uf); and 3) the exhaustive experiments demonstrating RePO's strong empirical performance and robustness against established baselines (noted by 1JuA, tpNX, uBRY).

In response to reviewer feedback, we conducted extensive new experiments during the rebuttal phase. These were well-received and led to score increases from Reviewers HXdp, M1uf, and uBRY. Key additions included:
* **Generalizability & Stability:** We integrated RePO's filtering into IPO (creating IPO-RePO), showing significant performance gains and demonstrating our principle's wide applicability. We also ran new high-$\beta$ experiments (up to $\beta=1000$) to empirically prove RePO's superior stability and performance over SimPO at its limit.
* **Scalability:** We added results on smaller models (Pythia-1.4B/2.8B), confirming RePO's effectiveness across different model sizes.

For the next version, we are committed to incorporating all promised revisions to fully address the reviewers' valuable suggestions. We will:
1.  **Integrate all new experiments:** The successful IPO-RePO, high-$\beta$ SimPO, and Pythia scalability results will be added to the main paper or appendix.
2.  **Strengthen over-optimization analysis:** We will add the requested KL-divergence comparison plots, further validating our claims.
3.  **Refine the narrative:** We will substantially revise the manuscript to sharpen the paper's focus on RePO as a "strong, simple, and theoretically-grounded baseline" and improve the tone and clarity as discussed.

We believe these comprehensive revisions will solidify our paper's contribution. Thank you for your consideration.

---

### Decision · Program_Chairs · 2025-09-17

**Decision:**

Accept (poster)

**Comment:**

In this paper, the authors propose a simple preference learning method, RePO, which uses ReLU activations. They theoretically show that RePO is a special case of SimPO (in the limit $\beta \to \infty$) and that its loss corresponds to the convex envelope of the 0/1 loss. RePO simplifies SimPO by eliminating the need to tune $\beta$, and it achieves competitive or superior performance compared with existing methods.

The reviewer and author discussion raised several issues, including the significance of the results, the coverage of ablations, and the scope of the experiments. After the discussion, all reviewers acknowledged the paper’s potential to some extent. The authors promised to integrate the new experiments provided during the discussion and to add experiments on overoptimization analysis, which satisfied many reviewers. Overall, although the idea is quite simple, the paper provides empirically well supported findings and offers insights of significant importance for language model alignment. In view of the above, I recommend acceptance.